# Two single-point mutations shift the ligand selectivity of a pheromone receptor between two closely related moth species

Ke Yang[1,2], Ling-Qiao Huang[1], Chao Ning[1,2], Chen-Zhu Wang[1,2]*

[1]State Key Laboratory of Integrated Management of Pest Insects and Rodents, Institute of Zoology, Chinese Academy of Sciences, Beijing, China; [2]College of Life Sciences, University of Chinese Academy of Sciences, Beijing, China

**Abstract** Male moths possess highly sensitive and selective olfactory systems that detect sex pheromones produced by their females. Pheromone receptors (PRs) play a key role in this process. The PR HassOr14b is found to be tuned to (Z)−9-hexadecenal, the major sex-pheromone component, in *Helicoverpa assulta*. HassOr14b is co-localized with HassOr6 or HassOr16 in two olfactory sensory neurons within the same sensilla. As HarmOr14b, the ortholog of HassOr14b in the closely related species *Helicoverpa armigera*, is tuned to another chemical (Z)−9-tetradecenal, we study the amino acid residues that determine their ligand selectivity. Two amino acids located in the intracellular domains F232I and T355I together determine the functional difference between the two orthologs. We conclude that species-specific changes in the tuning specificity of the PRs in the two *Helicoverpa* moth species could be achieved with just a few amino acid substitutions, which provides new insights into the evolution of closely related moth species.
DOI: https://doi.org/10.7554/eLife.29100.001

*For correspondence: czwang@ioz.ac.cn

Competing interests: The authors declare that no competing interests exist.

## Introduction

Almost all animals detect and react to pheromones and the other chemical cues that indicate food, shelter or predators, and their olfactory systems are mainly involved in the processes (*Wyatt, 2003*). As powerful chemical signals, pheromones are enormously varied in different animal species. How the animal olfaction has evolved at the molecular level to adapt to the changing pheromones is a forefront research subject in life sciences.

Moths are good model systems for pheromone communication study. Male moths fly upwind to find conspecific females releasing a plume of sex pheromone (*Cardé and Haynes, 2004*). Most moth sex pheromones have multiple components present in specific ratios that play significant roles in intraspecific sexual communication and in interspecific reproductive isolation (*Cardé et al., 1977*). Male moths possess highly sensitive and selective olfactory sensory neurons (OSNs) located in antennal sensilla that detect the pheromone molecules (*Schneider, 1964*; *Hansson and Stensmyr, 2011*). Pheromone receptors (PRs) located in the dendritic membrane of OSNs play a pivotal role in peripheral coding of sex pheromones (*Leal, 2013*; *Sakurai et al., 2004*).

Unlike general odorant receptors (ORs) that typically bind more than one ligand (*de Fouchier et al., 2017*), PRs are in general narrowly tuned to specific pheromone components (*Grosse-Wilde et al., 2007*; *Miura et al., 2010*; *Zhang and Löfstedt, 2015*). The ligands of some PRs in lepidopteran species have been successfully identified using heterologous expression systems, including *Xenopus* oocytes (*Wetzel et al., 2001*), the HEK293 cell line (*Grosse-Wilde et al., 2006*), the Sf9 cell line (*Kiely et al., 2007*), *Drosophila melanogaster delta-halo* mutants with an empty ab3A

neuron (*Dobritsa et al., 2003*), and *Or67D-GAL4* mutants (*Kurtovic et al., 2007*). However, the functions of many PRs in moth species are still unknown, thus hindering our understanding of pheromone detection at the molecular level in this group of insects.

Closely related moth species often use combinations of the same or similar pheromone components, which is a reflection of their common evolutionary history (*Cardé and Haynes, 2004*). They also possess homologous PRs with very similar sequences, but clearly differentiated in their ligand specificity. How do changes in amino acid sequences alter the ligand selectivity of PRs? The single study to date that has addressed this question showed that a single-point mutation in a PR is responsible for its different specificities in *Ostrinia furnacalis* and *Ostrinia nubilalis* (*Leary et al., 2012*). Studies of a number of ORs in *D. melanogaster* and *Anopheles gambiae* indicated that the determinant amino acids are located mainly in transmembrane domains and extracellular loops (*Guo and Kim, 2010*; *Hughes et al., 2014*; *Nichols and Luetje, 2010*; *Pellegrino et al., 2011*), but the molecular mechanisms that determine the ligand selectivity of ORs are still unclear.

The closely related moth species, *Helicoverpa assulta* and *Helicoverpa armigera*, are sympatric pests in Asia. The former is a specialist mainly feeding on solanaceous plants, including tobacco and hot pepper, whereas the latter is a polyphagous species and is one of the most devastating pests in the world. *H. assulta* and *H. armigera* share two compounds, (*Z*)−9-hexadecenal (Z9-16:Ald) and (*Z*)−11-hexadecenal (Z11-16:Ald) as their principal sex-pheromone components, but in inverse ratios, 93:7 and 3:97, respectively (*Piccardi et al., 1977*; *Wang et al., 2005*). (*Z*)−9-tetradecenal (Z9-14: Ald) acts as an antagonist in the pheromone communication of *H. assulta* (*Boo et al., 1995*; *Wu et al., 2015*). In that of *H. armigera*, Z9-14:Ald acts as an agonist in small amounts (0.3%) (*Rothschild, 1978*; *Wu et al., 2015*; *Zhang et al., 2012*) but an antagonist in higher amounts (1% and above) (*Gothilf et al., 1978*; *Kehat and Dunkelblum, 1990*; *Wu et al., 2015*). Three functional types of pheromone-sensitive sensilla, A, B and C, can be distinguished in the male antennae of the two species (*Baker et al., 2004*). Sensilla type A specifically respond to Z11-16:Ald, type B respond to Z9-14:Ald, and type C respond to Z9-16:Ald, Z9-14:Ald and some other structurally related compounds. B-type and C-type sensilla are classified into subtypes according to their response spectra (*Xu et al., 2016*). The population of A-type sensilla predominates in males of *H. armigera*, while C-type sensilla are most numerous in males of *H. assulta* (*Wu et al., 2013*; *Xu et al., 2016*). The two species share almost the same set of orthologous PRs. Previous functional studies of the PRs showed that HarmOr13 and HassOr13 are specifically tuned to Z11-16:Ald (*Jiang et al., 2014*; *Liu et al., 2013*), HarmOr14b and HassOr16 are tuned to Z9-14:Ald (*Jiang et al., 2014*; *Liu et al., 2013*), HarmOr16 is tuned to both Z9-14:Ald and (*Z*)−11-hexadecenol (Z11-16:OH) (*Liu et al., 2013*), while HarmOr6 and HassOr6 are mainly tuned to (*Z*)−9-hexadecenol (Z9-16:OH) (*Jiang et al., 2014*). However, it is still unclear which PR is specific for Z9-16:Ald, the major component of *H. assulta* sex pheromone.

In this study, we first identified the PR tuned to Z9-16:Ald in *H. assulta*. Because C-type sensilla responding to Z9-16:Ald are densely distributed in the male antennae of *H. assulta*, we predicted that this PR should be highly expressed in male antennae. Therefore, we used qPCR to analyze the expression level of all candidate PRs in male antennae in *H. assulta*, and then used the *Xenopus* oocyte expression system and two-electrode voltage-clamp recording to examine the function of highly expressed PRs. We surprisingly found that the PR tuned to Z9-16:Ald is HassOr14b, while its ortholog HarmOr14b is tuned to Z9-14:Ald in the closely related species *H. armigera*. Next, focusing on the two orthologous receptors, we identified the amino acid residues determining this functional shift. We used a series of regional replacements and single-point mutations, coupled with functional analyses, to demonstrate that two single-point mutations located in the intracellular regions of the molecule together determine their ligand selectivity. Our results suggest that a change in the tuning selectivity of PRs during the speciation of some moths could result from just a few mutations.

## Results

### Phylogenetic analysis of candidate PRs

The reported transcriptome data and full-length cloning of the PRs made it possible to analyze all candidate PRs in the two closely related species and in other species of Noctuidae. The amino acid sequences of seven PRs from *Helicoverpa* species (*Jiang et al., 2014*; *Liu et al., 2014*; *Xu et al.,*

*2015*) and 32 PRs from other noctuids were used to construct a phylogenetic tree, where the Orco sequences represented an outgroup (*Krieger et al., 2004*; *Liu et al., 2013*; *Mitsuno et al., 2008*; *Montagné et al., 2012*; *Zhang and Löfstedt, 2013*; *Zhang et al., 2015*, Zhang et al., 2014Zhang et al., 2014Zhang et al., 2014*Zhang et al., 2014*) (*Figure 1*). In general, the tree was clustered into seven lineages, Or16, Or6, Or14b, Or14, Or15, Or11, and Or13. Each lineage contains the PR(s) from *Helicoverpa* and other noctuids except for Or14b, suggesting that the Or14b cluster specifically occurs in *H. assulta* and *H. armigera*. To investigate the evolutionary pressures acting on the coding regions of each cluster, we estimated the ratios of nonsynonymous (dN) to synonymous (dS) nucleotide substitution ($\omega$ = dN/dS) in the PR gene lineages and the Orco lineage using DnaSP version 5.10 (*Librado and Rozas, 2009*). The $\omega$ values < 1 were observed in all PR clusters and the Orco cluster (cluster Or16: $\omega$ <0.21; cluster Or6: $\omega$ <0.19; cluster Or14b: $\omega$ = 0.17; cluster Or14: $\omega$ <0.15; cluster Or15: $\omega$ = 0.13; cluster Or11: $\omega$ <0.13; cluster Or13, $\omega$ <0.17; cluster Orco: $\omega$ <0.03); this indicates that all the PRs and Orcos analyzed in this study are subjected to purifying selection, which is consistent with the previous studies (*Zhang and Löfstedt, 2013*; *Zhang et al., 2014*).

## Expression level of candidate PRs in antennae of *H. assulta* and *H. armigera*

The antennal expression levels in males and females were compared using quantitative real-time PCR (qPCR). All the candidate PRs were male-specific except for Or11, which was highly expressed in both male and female antennae (*Figure 2* and *Figure 2—figure supplement 2*). In the male antennae of *H. assulta*, HassOr14b had the highest expression level, nearly twofold higher than the levels of HassOr6 and HassOr16, and five- to sixfold greater than that of HassOr13 (*Figure 2*). The values of fragments per kilobase of transcript per million reads (FPKM) in different tissues of *H. assulta* further demonstrated that HassOr14b is specifically expressed in the male antennae and that its expression level is the highest among the PRs (*Figure 2—figure supplement 1*). In the male antennae of *H. armigera*, HarmOr13 and HarmOr11 showed the highest expression level, which was about five- to sixfold higher than HarmOr16 and HarmOr14; HarmOr14b had a low expression level, even lower than HarmOr16 and HarmOr6 (*Figure 2—figure supplement 2*). Since the C-type sensilla responding to Z9-16:Ald were the most abundant type in the male antennae of *H. assulta*, we speculated that HassOr14b would be the PR tuned to Z9-16:Ald, different from HarmOr14b which tuned to Z9-14:Ald.

## PR specifically tuned to Z9-16:Ald in *H. assulta*

We re-cloned the sequence of HassOr14b and verified it by Sanger sequencing and the transcriptome data of *H. assulta*. We used the *Xenopus laevis* oocyte expression system and two-electrode voltage-clamp recording to study the function of HassOr14b although it has already been shown that its ortholog, HarmOr14b, is tuned to Z9-14:Ald (*Jiang et al., 2014*). Oocytes expressing HassOr14b/HassOrco responded robustly to Z9-16:Ald, and to a much lesser extent to Z9-16:OH at a concentration of $10^{-4}$ M (*Figure 3A*). Z9-16:Ald induced currents increasing from the lowest threshold concentration of $10^{-6}$ M to $3.3 \times 10^{-3}$ M in a dose-dependent manner with an $EC_{50}$ value of $8.65 \times 10^{-5}$ M (*Figure 4*). We also verified the function of the ortholog of HassOr14b, HarmOr14b and the next most highly expressed PRs in male *H. assulta*, HassOr6 and HassOr16. As previously reported (*Jiang et al., 2014*), we verified that HarmOr14b is specifically tuned to Z9-14:Ald, and also weakly responds to Z9-16:Ald (*Figure 3B*), while HassOr6 is mainly tuned to Z9-16:OH (*Figure 3—figure supplement 1A*), and HassOr16 is specific for Z9-14:Ald (*Figure 3—figure supplement 1B*). Water-injected oocytes fail to respond to any of the pheromone component stimuli as negative controls (*Figure 3—figure supplement 2* and *Figure 4—figure supplement 1*).

## Co-localization of HassOr14b with other PRs in type C sensilla

Based on the above results and previous reports (*Jiang et al., 2014*; *Xu et al., 2016*), we considered that HassOr14b, HassOr6 and HassOr16 were most likely to be the PRs expressed in C-type sensilla of *H. assulta*. By two-color *in situ* hybridization, we further analyzed the co-localization of these three PRs. We found that HassOr14b and HassOr6 were co-localized in some sensilla (arrows, *Figure 5A1–4*), while in other sensilla only HassOr14b was detected (arrows, *Figure 5B1–4*). A similar situation was observed for HassOr14b and HassOr16. They were co-localized in some sensilla

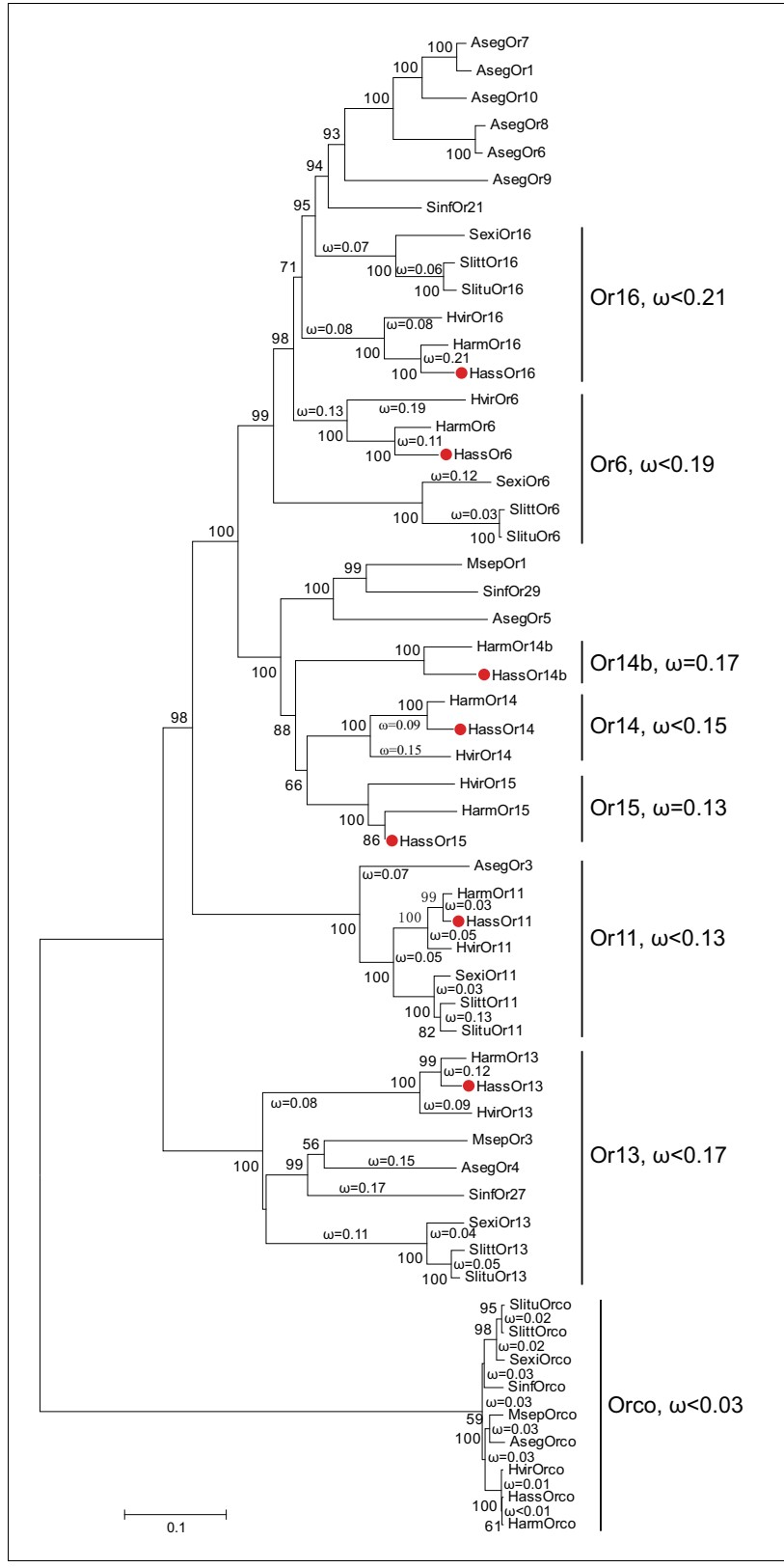

**Figure 1.** The phylogenetic tree of the PRs in Noctuidae. The amino acid sequences are based on the reported transcriptome data of functionally identified PRs. The Orco lineage is defined as an outgroup. Bootstrap values are based on 1000 replicates, and values over 50 are shown at corresponding nodes. The bar indicates the phylogenetic distance value. The nonsynonymous (dN) to synonymous (dS) substitution ratio (ω) is labeled in the

*Figure 1 continued on next page*

*Figure 1 continued*

tree. Cluster Or14b and Or15 have a uniform ω value for all branches, whereas Cluster Or6, 11, 13, 14, and 16 have varying ω values for all branches within the lineage. The ω values of all clusters were less than 1, suggesting that all PRs were subjected to purifying selection. Abbreviations: Aseg, *Agrotis segetum*; Harm, *H. armigera*; Hass, *H. assulta*; Hvir, *Heliothis virescens*; Msep, *Mythimna separata*; Sexi, *Spodoptera exigua*; Slitt, *Spodoptera littoralis*; Slitu, *Spodoptera litura*; Sinf, *Sesamia inferens*. The PRs of *H. assulta* are indicated by red dots '•'. The GenBank accession numbers of genes used in this analysis are listed in *Supplementary file 2*.
DOI: https://doi.org/10.7554/eLife.29100.002

(arrows, *Figure 5C1–4*), but only HassOr14b was detected in other sensilla (arrows, *Figure 5D1–4*). However, HassOr6 and HassOr16 were always expressed in different sensilla (arrows, *Figure 5E1–4*). These results indicate that HassOr14b is co-localized with HassOr6 or HassOr16 in different C type sensilla.

## Regional mutations of HassOr14b and functional analysis

HassOr14b and HarmOr14b exhibit 91% amino acid identity (402 out of 440, *Figure 6—figure supplement 1*), but their ligand selectivity is different. This provided an opportunity to examine the relationship between structure and function in the two orthologous PRs. From the sequence alignment and secondary structural analysis (*Figure 6*; TOPCONS, topcons.net), we found that the 38 differing amino acids were distributed fairly uniformly in the two proteins. Therefore, we separated the whole sequence into eight regions (RI–VIII) (*Figure 6*). Then we conducted a series of mutagenesis experiments by replacing each of the eight regions of HassOr14b with the corresponding segment of HarmOr14b, while maintaining the rest of the sequence unchanged. After successfully constructing the modified sequences, we analyzed their functions as for the wild type (*Figure 7*). Interestingly, comparing with the ligand selectivity of the wild type (*Figure 7—figure supplement 2*), we observed that the ligand selectivity of HassOr14b was changed remarkably by replacement of the region VI or VIII. HassOr14b after replacing the region VI had a significantly higher response to Z9-14:Ald than to Z9-16:Ald (*Figure 7F*), while after replacing the region VIII had strong responses to both Z9-16:Ald and Z9-14:Ald (*Figure 7H*). However, most of the region replacements in HassOr14b showed selectivities similar to that of the wild type, with Z9-16:Ald being the most effective ligand. In particular, replacement of the regions I or III produced significantly lower responses to Z9-16:Ald (*Figure 7A and C*), while replacement of the region VII resulted in a significantly stronger response to Z9-16:Ald (*Figure 7G*). Replacement of the regions II, IV or V did not affect the selectivity of the receptor with reference to the wild-type (*Figure 7B,D and E*).

## Site-specific mutations of HassOr14b and functional analysis

Based on the observation that the ligand selectivity of HassOr14b was changed only by replacement of the region VI or VIII, we chose these two segments of the receptor for further single-site mutations and functional analysis. Five amino acids (E188G, E196D, F232I, R262K, and R270K) in the region VI, and three (T355I, R395K, and A425K) in the region VIII were different between HassOr14b and HarmOr14b (*Figures 6* and *8*). We replaced each amino acid in

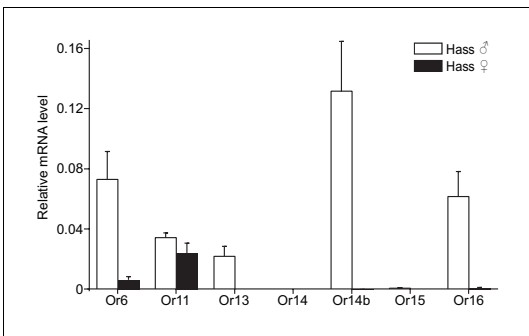

**Figure 2.** Relative mRNA expression levels of PRs by quantitative real-time PCR analysis in male and female antennae of *H. assulta*. Hass♂, male antennae; Hass♀, female antennae. *n* = 3 replicates of 40–60 antennae each. Data are presented as mean ± SEM.
DOI: https://doi.org/10.7554/eLife.29100.003

The following figure supplements are available for figure 2:

**Figure supplement 1.** The tissue expression pattern of PRs in *Helicoverpa assulta* by Illumina read-mapping analysis.
DOI: https://doi.org/10.7554/eLife.29100.004

**Figure supplement 2.** Relative mRNA expression levels of PRs by quantitative real-time PCR analysis in male and female antennae of *Helicoverpa armigera*.
DOI: https://doi.org/10.7554/eLife.29100.005

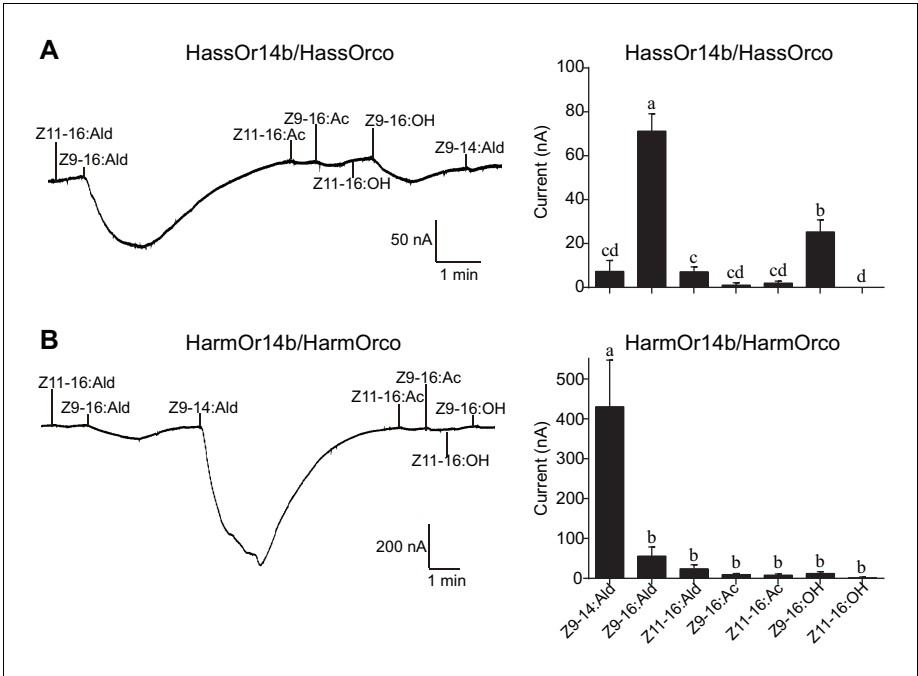

**Figure 3.** Two-electrode voltage-clamp recordings of *Xenopus* oocytes with co-expressed HassOr14b/HassOrco, and HarmOr14b/HarmOrco, stimulated with pheromone components and analogs. (A) Inward current responses (left) and response profiles (right) of *Xenopus* oocytes expressing HassOr14b/HassOrco in response to $10^{-4}$ M concentrations of pheromone components and analogs. $n = 7$ replicates of cells, $F = 31.75$, p<0.001, one-way ANOVA, Tukey HSD test. (B) Inward current responses (left) and response profiles (right) of *Xenopus* oocytes expressing HarmOr14b/HarmOrco in response to $10^{-4}$ M concentrations of pheromone components and analogs. $n = 7$ replicates of cells, $F = 17.67$, p<0.001, one-way ANOVA, Tukey HSD test. Data are presented as mean ± SEM.

DOI: https://doi.org/10.7554/eLife.29100.006

The following figure supplements are available for figure 3:

**Figure supplement 1.** Two-electrode voltage-clamp recordings of *Xenopus* oocytes with co-expressed HassOr6/HassOrco and HassOr16/HassOrco to stimulation with pheromone compounds and analogs.

DOI: https://doi.org/10.7554/eLife.29100.007

**Figure supplement 2.** Two-electrode voltage-clamp recordings of *Xenopus* oocytes injected with distilled water and stimulated with pheromone compounds and analogs.

DOI: https://doi.org/10.7554/eLife.29100.008

**Figure supplement 3.** Alignment of amino acid sequences of HassOr14b in three studies: Yang et al.

DOI: https://doi.org/10.7554/eLife.29100.009

turn by mutating each of the eight residues. Comparing with the wild type (*Figure 7—figure supplement 2*), we found that the mutant F232I was activated more by Z9-14:Ald than by Z9-16:Ald (*Figure 8C*), while the mutant T355I showed very strong responses to both Z9-16:Ald and Z9-14:Ald (*Figure 8F*). The other mutations showed largely the same selectivity as the wild type although with different values of the currents. Compared to the wild type, E196D and R262K still responded to Z9-16:Ald but showed a decrease in current (*Figure 8B and D*), E188G and R270K also responded to Z9-16:Ald but showed an increase in current (*Figure 8A and E*), while R395K and A425K exhibited the same response level to Z9-16:Ald and, to a minor extent, to Z9-14:Ald (*Figure 8G and H*).

We next constructed a mutant bearing the two substitutions (F232I and T355I) that affected the ligand selectivity of HassOr14b. This two-site mutant showed a robust response to Z9-14:Ald and a minor response to Z9-16:Ald, reproducing the characteristic selectivity of HarmOr14b (*Figure 8I*).

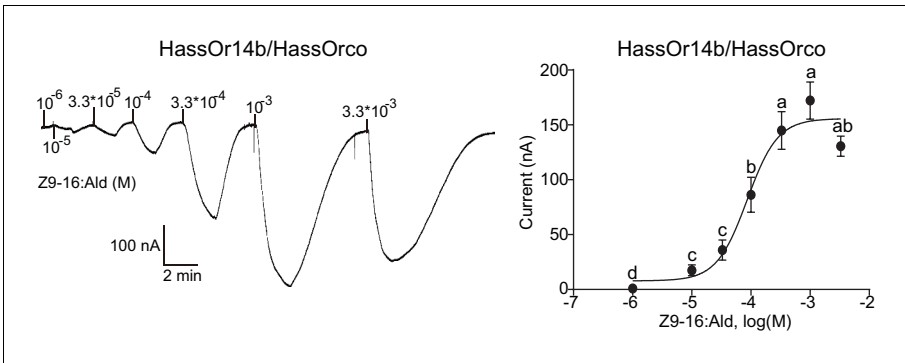

**Figure 4.** Dose responses of *Xenopus* oocytes with co-expressed HassOr14b/HassOrco stimulated with a range of Z9-16:Ald concentrations. Inward current responses (left) and response profiles (right) of *Xenopus* oocytes expressing HassOr14b/HassOrco in response to Z9-16:Ald at serial concentrations. The EC$_{50}$ value for Z9-16:Ald was $8.65 \times 10^{-5}$ M. $n$ = 7–9 replicates of cells, $F$ = 54.57, p<0.001, one-way ANOVA, Tukey HSD test.
DOI: https://doi.org/10.7554/eLife.29100.010

The following figure supplement is available for figure 4:

**Figure supplement 1.** Two-electrode voltage-clamp recordings of *Xenopus* oocytes injected with distilled water and stimulated with pheromone compounds and analogs.
DOI: https://doi.org/10.7554/eLife.29100.011

## Discussion

Characteristics of chemosensory receptor binding sites are emerging for vertebrate ORs, which are seven transmembrane-spanning G-protein-coupled receptors, but less is known about insect ORs (*Kato and Touhara, 2009*; *Ramdya and Benton, 2010*). In this study, we identify HassOr14b as the PR tuned to Z9-16:Ald, the major sex-pheromone component of *H. assulta*. Its ortholog HarmOr14b is specific for Z9-14:Ald in *H. armigera* and we further demonstrate that two single-point mutations, F232I and T355I, located in the intracellular domains of the receptor, together determine the functional shift between orthologs in the two closely related species.

### Novel identified PR and the different combinations with other PRs

As the number of different kinds of OSNs in three types of sensilla is related to the expression level of the corresponding PRs, the characteristics and abundance of the sensilla thus can provide reliable information for identifying PRs' function. The previous studies clarified that the C type sensilla responding to Z9-16:Ald are predominant in male antennae of *H. assulta* (*Wu et al., 2013*; *Wu et al., 2015*; *Xu et al., 2016*), the PR tuning to Z9-16:Ald must be highly expressed in male antennae. We compare the expression level of all candidate PR genes in the male antennae of *H. assulta* and *H. armigera*. We found that HassOr14b is the most highly expressed in *H. assulta*, while HarmOr14b has relatively low expression level in *H. armigera*. The functional study confirms that HassOr14b is specifically tuned to Z9-16:Ald, while its ortholog HarmOr14b is specifically tuned to Z9-14:Ald. This suggests that the two closely related species not only changed Or14b's expressing level, but also altered its tuning selectivity. It is worth noting that inward currents of the oocytes expressing HassOr14b induced by Z9-16:Ald were distinct but relatively low. It is common that the oocytes expressing some PRs are relatively weaker than others in responding to their ligands. However, their responding patterns in the oocyte system are generally representative of those in native OSNs. A clear dose-response curve to the most effective ligand is always helpful to confirm the receptor's function.

The previous functional studies of HassOr14b did not find its activity by using the *Xenopus* system (*Chang et al., 2016*; *Jiang et al., 2014*). In this study, we re-cloned the sequence of *HassOr14b* by use of Q5 High-Fidelity DNA Polymerase (New England Biolabs, Ipswich, MA) and repeated again to verify the sequence by Sanger sequencing for 10 samples, and also compared with the sequence in the transcriptome data of *H. assulta*. Finally, we got the correct sequence, in which there are three amino acids different from the sequence in *Jiang et al. (2014)* (*Figure 3—figure supplement 3*).

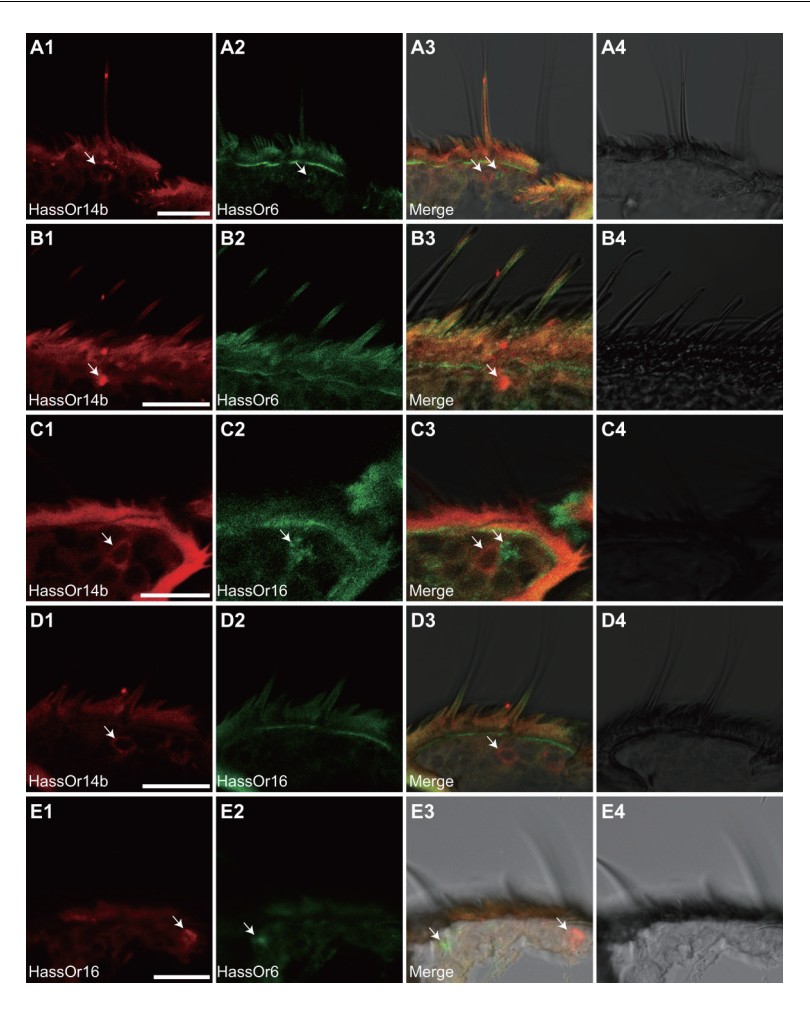

**Figure 5.** Two-colour *in situ* hybridization visualizing the combinations of HassOr14b/HassOr6, HassOr14b/HassOr16, and HassOr6/HassOr16 in male antennae of *H.assulta*. (**A, B**) The localization of HassOr14b and HassOr6. (**C, D**) The localization of HassOr14b and HassOr16. (**E**) The localization of HassOr6 and HassOr16. Signals were visualized by red (digoxin-labeled probes) (**A1, B1, C1, D1**), green (biotin-labeled probes) (**A2, B2, C2, D2**), and both red and green (**A3, B3, C3, D3**) fluorescence. Bright-field images are presented as references (**A4, B4, C4, D4**). Arrows indicate the cell location. Scale bars: 20 μm.
DOI: https://doi.org/10.7554/eLife.29100.012

Moreover, we further analyzed the transcriptome data in *H. assulta* and confirmed that in the three different amino acids positions, there is no sequence polymorphism. We used the accurate sequence this time and characterized the function of HassOr14b, which is specifically tuned to Z9-16:Ald, the major sex pheromone component in *H. assulta*. *Chang et al., 2016* also used the LA-*Taq* polymerase (TaKaRa, Shiga, Japan) *Jiang et al. (2014)* used before when they cloned the sequence, and there is one amino acid different in the 5' ends from ours (*Figure 3—figure supplement 3*). By analyzing the transcriptome data in *H. assulta,* we confirmed that this amino acid position has no sequence polymorphism. Another difference between the two studies is the vector used in the expression system. *Chang et al., 2016* used the pT7Ts vector, while we use the pCS2+ vector in the expression system. We suggest that the accuracy and integrity of the sequence is crucial to identify the function of the receptors. Moreover, the selection of the appropriate expression vector could be also important.

As the ligands of HassOr14b and the second abundant PRs, HassOr6 and HassOR16 are all included in the responding spectrum of the C type sensilla, we further investigated the expressing sites of the three PRs in the sensilla. HassOr14b is co-localized with HassOR6 or HassOR16 in the

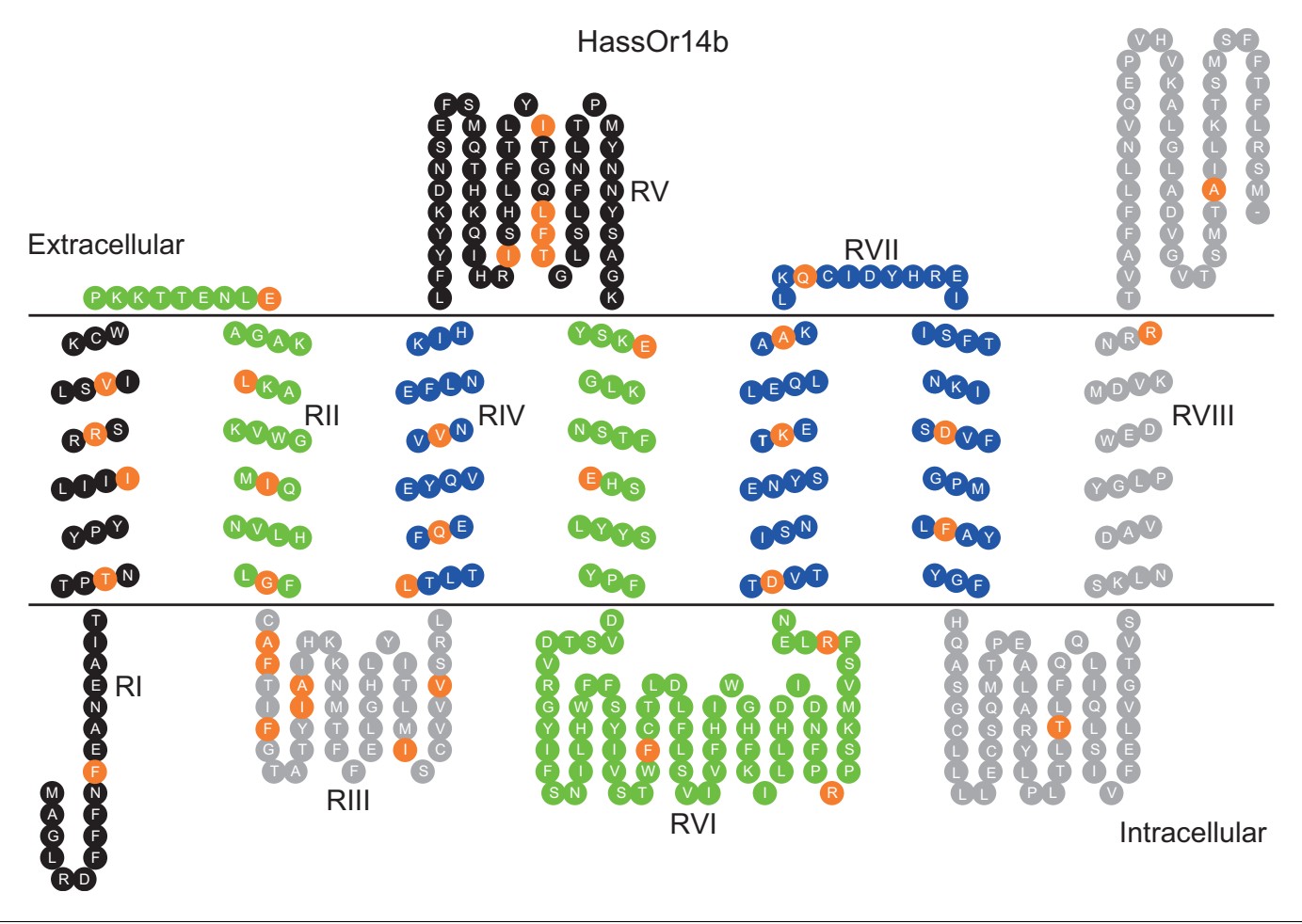

**Figure 6.** Eight mutation regions in the predicted secondary structure of HassOr14b. Each circle indicates an amino acid residue that differs between HassOr14b and HarmOr14b and is highlighted in orange. Black indicates mutation regions RI and RV; green indicates RII and RVI; grey indicates RIII and RVIII; blue indicates RIV and RVII. The image was constructed by TOPO2 software (http://www.sacs.ucsf.edu/TOPO2/) based on the secondary structure predicted by TOPCONS (topcons.net) models (*Tsirigos et al., 2015*). The structures of both HassOr14b and HarmOr14b were predicted and the model with a reliable 7-transmembrane structure was adopted.

DOI: https://doi.org/10.7554/eLife.29100.013

The following figure supplement is available for figure 6:

**Figure supplement 1.** Alignment of amino acid sequences of HassOr14b and HarmOr14b.

DOI: https://doi.org/10.7554/eLife.29100.014

neighboring neurons in the same sensilla, while HassOr6 and HassOr16 are always expressed in different sensilla, which is different from the previous study (*Chang et al., 2016*). Our results indicate that there are different combinations of the PRs in the C type sensilla, which is consistent with the previous single sensillum recording results that there are subtypes in the type C sensilla (*Xu et al., 2016*). To the best of our knowledge, this is the first study that shows the various combinations of PRs were the molecular basis for the different sensilla subtypes in moth species.

## Two amino acids located in the intracellular domains together determine the OR selectivity

Insects use olfactory receptors to discriminate amongst thousands of volatiles or pheromones (*Kaupp, 2010*). Insect ORs require the co-expression of a ligand-selective OR and a universal odorant co-receptor (Orco) to form ligand-gated ion channels (*Missbach et al., 2014*; *Vosshall and Hansson, 2011*). In the absence of data on the crystalline structure of insect ORs, the relationship

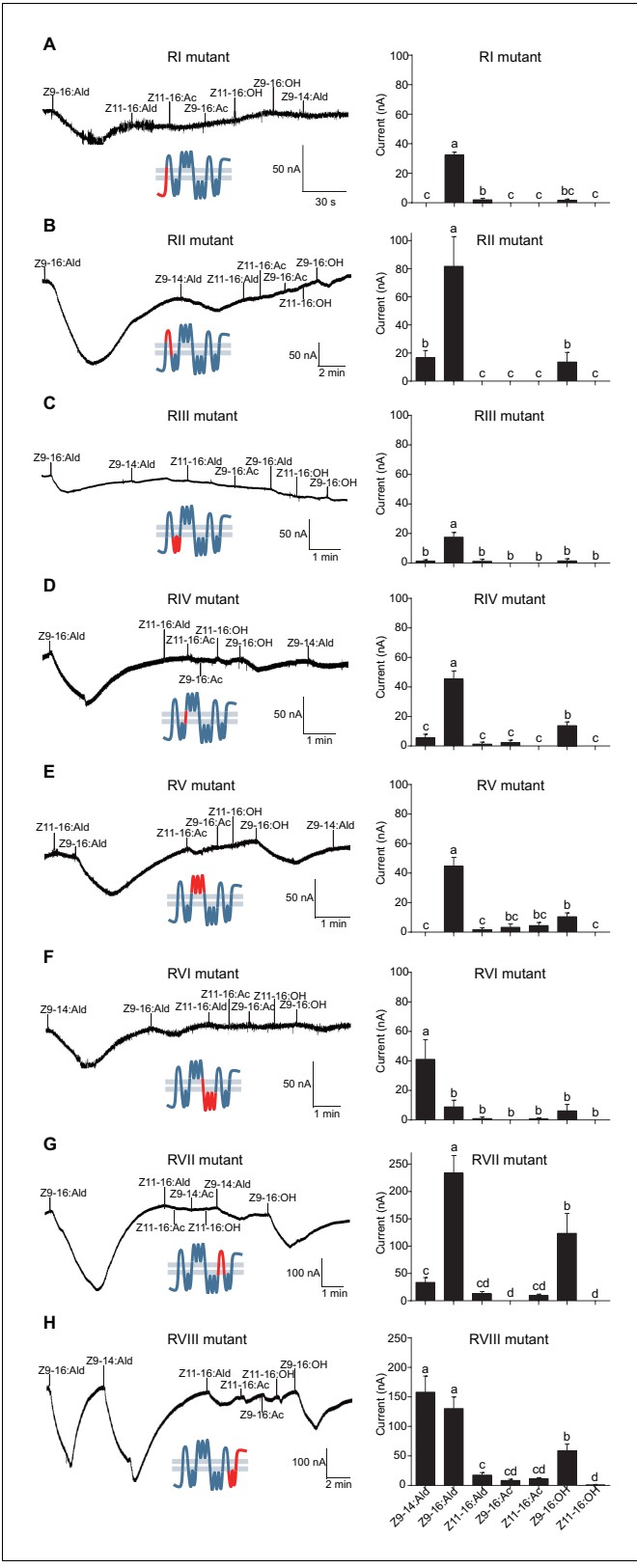

**Figure 7.** Two-electrode voltage-clamp recordings of *Xenopus* oocytes with co-expressed regional mutations and HassOrco stimulated with pheromone components and analogs. (**A**) Inward current responses (left) and response profiles (right) of *Xenopus* oocytes expressing RI mutant/HassOrco in response to $10^{-4}$ M concentrations of pheromone components and analogs. $n$ = 7 replicates of cells, $F$ = 85.28, p<0.001. (**B**) Inward current responses

*Figure 7 continued on next page*

*Figure 7 continued*

(left) and response profiles (right) of *Xenopus* oocytes expressing RII mutant/HassOrco in response to $10^{-4}$ M concentrations of pheromone components and analogs. $n = 6$ replicates of cells, $F = 29.11$, p<0.001. (**C**) Inward current responses (left) and response profiles (right) of *Xenopus* oocytes expressing RIII mutant/HassOrco in response to $10^{-4}$ M concentrations of pheromone components and analogs. $n = 5$ replicates of cells, $F = 24.94$, p<0.001. (**D**) Inward current responses (left) and response profiles (right) of *Xenopus* oocytes expressing RIV mutant/HassOrco in response to $10^{-4}$ M concentrations of pheromone components and analogs. $n = 6$ replicates of cells, $F = 31.12$, p<0.001. (**E**) Inward current responses (left) and response profiles (right) of *Xenopus* oocytes expressing RV mutant/HassOrco in response to $10^{-4}$ M concentrations of pheromone components and analogs. $n = 6$ replicates of cells, $F = 23.80$, p<0.001. (**F**) Inward current responses (left) and response profiles (right) of *Xenopus* oocytes expressing RVI mutant/HassOrco in response to $10^{-4}$ M concentrations of pheromone components and analogs. $n = 7$ replicates of cells, $F = 14.77$, p<0.001. (**G**) Inward current responses (left) and response profiles (right) of *Xenopus* oocytes expressing RVII mutant/HassOrco in response to $10^{-4}$ M concentrations of pheromone components and analogs. $n = 6$ replicates of cells, $F = 39.40$, p<0.001. (**H**) Inward current responses (left) and response profiles (right) of *Xenopus* oocytes expressing RVIII mutant/HassOrco in response to $10^{-4}$ M concentrations of pheromone components and analogs. $n = 10$ replicates of cells, $F = 41.57$, p<0.001. Data are presented as mean ± SEM. One-way ANOVA, Tukey HSD test are used. Mutation regions are highlighted in red.

DOI: https://doi.org/10.7554/eLife.29100.015

The following figure supplements are available for figure 7:

**Figure supplement 1.** The construction strategy of HassOr14b mutation sequences.
DOI: https://doi.org/10.7554/eLife.29100.016

**Figure supplement 2.** Two-electrode voltage-clamp recordings of *Xenopus* oocytes with co-expressed wild-type HassOr14b/HassOrco, and wild type HarmOr14b/HarmOrco, stimulated with pheromone components and analogs.
DOI: https://doi.org/10.7554/eLife.29100.017

between structure and function in these molecules is elusive. By amino acid covariation across insect Orcos and ORs, Hopf *et al*. constructed the first 3D models of *D. melanogaster* ORs (*Hopf et al., 2015*). However, this provided only an indirect insight into protein structure (*Carraher et al., 2015*). Previous site-directed mutagenesis studies performed to probe OR specificity, mainly focused on the transmembrane domains (TMDs) and extracellular loops (ECLs), based on the assumption that the TMDs and ECLs of the OR form the ligand-binding pocket (*Guo and Kim, 2010*). Leary *et al*. reported that a single amino acid mutation located in the predicted third TMD could change the ligand specificity of a PR between that of the Asian corn borer and that of the European corn borer (*Leary et al., 2012*). Pellegrino *et al*. showed that a single natural polymorphism of *D. melanogaster* Or59B in the third transmembrane domain altered DEET sensitivity (*Pellegrino et al., 2011*). In *A. gambiae*, a single mutation of AgOr15 at the interface between ECL2 and TMD4, produced large changes in responses to odors (*Hughes et al., 2014*). To address the relationship between OR-Orco structure and function, several recent studies showed that some amino acid residues in the OR or Orco were essential for channel activity of the heteromeric insect OR-Orco complex (*Kumar et al., 2013*; *Nakagawa et al., 2012*; *Turner et al., 2014*).

The different ligand selectivities of HassOr14b and HarmOr14b provide a convenient system in which to study structure-function relationships of PRs. Comparing the whole amino acid sequences of the two orthologous receptors, we identified two regions that were responsible for their selectivity. This new method is convenient and efficient, particularly for functional comparisons between orthologous or paralogous genes with many differing amino acids. By further replacing single amino acids in the two regions, we finally detected two single-point mutations, T355I and F232I responsible for the different ligand selectivities of HassOr14b and HarmOr14b. It is for the first time to find that the two mutation sites in the intracellular domains (ICDs) rather than in the TMDs and ECLs were involved in determination of ligand selectivity. We suggest two possible explanations for the role of ICDs. First, the binding site of ligand-specific ORs, such as PRs, may have a complex structure, which involves TMDs (*Leary et al., 2012*), ECLs (*Hughes et al., 2014*) and ICDs. Alternatively, ICDs may be involved in the specific interactions of the PR with the related G proteins. To relay the signal into the cell interior, binding of an extracellular molecule to an OR is tightly followed by binding of the receptor to a trimeric G protein inside the cell (*Ignatious Raja et al., 2014*; *Wicher et al., 2008*).

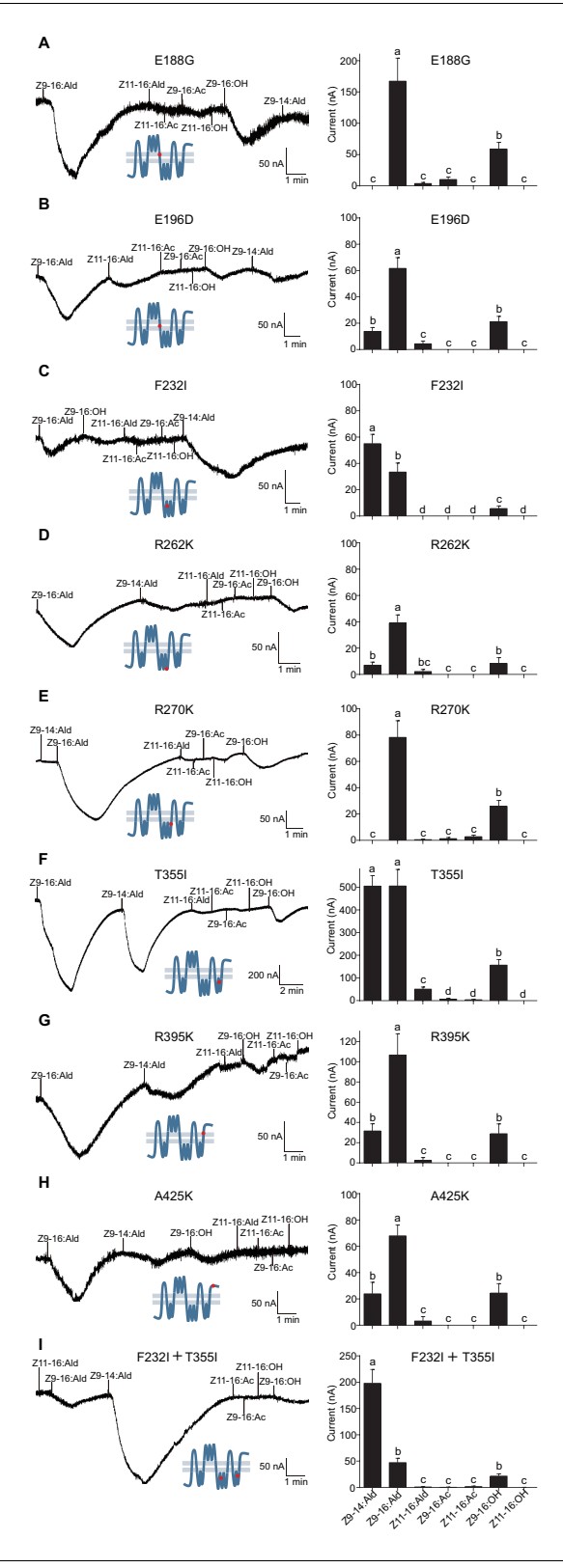

**Figure 8.** Two-electrode voltage-clamp recordings of *Xenopus* oocytes with co-expressed site mutations and HassOrco, stimulated with pheromone components and analogs. (A) Inward current responses (left) and response profiles (right) of *Xenopus* oocytes expressing E188G/HassOrco in response to $10^{-4}$ M concentrations of pheromone components and analogs. $n$ = 6 replicates of cells, $F$ = 52.32, p<0.001. (B) Inward current responses

*Figure 8 continued on next page*

*Figure 8 continued*

(left) and response profiles (right) of *Xenopus* oocytes expressing E196D/HassOrco in response to $10^{-4}$ M concentrations of pheromone components and analogs. $n = 7$ replicates of cells, $F = 65.87$, p<0.001. (**C**) Inward current responses (left) and response profiles (right) of *Xenopus* oocytes expressing F232I/HassOrco in response to $10^{-4}$ M concentrations of pheromone components and analogs. $n = 8$ replicates of cells, $F = 90.65$, p<0.001. (**D**) Inward current responses (left) and response profiles (right) of *Xenopus* oocytes expressing R262K/HassOrco in response to $10^{-4}$ M concentrations of pheromone components and analogs. $n = 8$ replicates of cells, $F = 31.96$, p<0.001. (**E**) Inward current responses (left) and response profiles (right) of *Xenopus* oocytes expressing R270K/HassOrco in response to $10^{-4}$ M concentrations of pheromone components and analogs. $n = 8$ replicates of cells, $F = 56.13$, p<0.001. (**F**) Inward current responses (left) and response profiles (right) of *Xenopus* oocytes expressing T355I/HassOrco in response to $10^{-4}$ M concentrations of pheromone components and analogs. $n = 9$ replicates of cells, $F = 85.70$, p<0.001. (**G**) Inward current responses (left) and response profiles (right) of *Xenopus* oocytes expressing R395K/HassOrco in response to $10^{-4}$ M concentrations of pheromone components and analogs. $n = 7$ replicates of cells, $F = 40.19$, p<0.001. (**H**) Inward current responses (left) and response profiles (right) of *Xenopus* oocytes expressing A425K/HassOrco in response to $10^{-4}$ M concentrations of pheromone components and analogs. $n = 5$ replicates of cells, $F = 24.24$, p<0.001. (**I**) Inward current responses (left) and response profiles (right) of *Xenopus* oocytes expressing (F232I + T355I)/HassOrco in response to $10^{-4}$ M concentrations of pheromone components and analogs. $n = 9$ replicates of cells, $F = 79.03$, p<0.001. Data are presented as mean ± SEM. One-way ANOVA, Tukey HSD test are used. Mutation sites are highlighted by red dots '•'.

DOI: https://doi.org/10.7554/eLife.29100.018

Elucidation of the details of the structural and functional mechanisms of ORs must await further study.

## Implications for the modulation and evolution of OR selectivity

Animal nervous systems are shaped by shifting environmental selection pressures to perceive and respond to new sensory cues (*Prieto-Godino et al., 2017*). The olfactory systems found in all animals have nearly the same design features, which give olfaction a considerable flexibility for signaling to evolve. Since the central odor processing is relatively conserved, new olfactory pathways tend to evolve from the peripheral changes (*Galizia and Rössler, 2010*; *Prieto-Godino et al., 2017*). How mutations in olfactory receptors change the olfactory responses of animals and eventually impact on the evolution of animal behavior is crucial but remains unclear.

In moth species, the co-evolution of pheromones produced by females and their detection by males present a paradox. Under stabilizing selection, variation of the female pheromone blend is limited, and the males typically prefer the most common pheromone blends (*Groot et al., 2016*; *Roelofs et al., 2002*). The ω value for all clusters of PRs analyzed in this study are less than 1, suggesting that PRs would be subjected to purifying selection. However, at the same time, the male moths need to have a degree of plasticity to adapt to changes in signal structures associated with speciation. Site-directed mutagenesis and functional analyses could validate how many amino acid substitutions are required to alter a PR's selectivity.

Based on previous studies, suggesting that *H. assulta* is ancestral to *H. armigera* (*Cho et al., 2008*; *Fang et al., 1997*), we tried to reproduce the assumed evolutionary process that mutated HassOr14b into HarmOr14b. Most of the regional replacements and site mutations did not change the ligand selectivity of HassOr14b, indicating the functional stability of this PR. Only F232I and T355I substitutions produced a large change of the ligand selectivity of Or14b, from Z9-16:Ald in *H. assulta* to Z9-14:Ald in *H. armigera*. The former site mutation produced a small shift from Z9-16:Ald to Z9-14:Ald in the response spectrum, the latter extended and strengthened the responses to both chemicals, while the two mutations together generated a complete functional shift from Z9-16:Ald to Z9-14:Ald. These results indicate that the substitutions of a few key amino acids are able to greatly change PR selectivity, laying the molecular foundations for PR plasticity. Moreover, it seems that at least two steps, involving in the functional extension and shift, are required for a major functional change of HassOr14b, each step with a single point mutation. In the course of speciation, the functional change of ORs could be a process with multiple amino acid mutations, a few making drastic changes and many making small modifications or even no change in function.

The two closely related species *H. assulta* and *H. armigera* are one of the ideal study systems for pheromone communication. They share two chemicals, Z9-16:Ald and Z11-16:Ald, as their principal

sex pheromone components but with reverse ratios. Males possess sensitive olfactory systems to detect conspecific sex pheromone blends. Peripheral coding of the binary blends with reversed ratios is mainly attributed to two group of specific OSNs in separate antennal sensilla with reverse population sizes, which reflect different expression levels of related PRs (*Wu et al., 2013*). In this study, we discover that HassOr14b, the highest expressed PR in male antennae of *H. assulta* is tuned to Z9-16:Ald, the major component of the sex pheromone of *H. assulta*, while its ortholog HarmOr14b is tuned to Z9-14:Ald in *H. armigera*, which provides us an ideal model to study the determinants of OR selectivity. We systematically identify two single-point mutations, F232I and T355I, located in the intracellular regions of HassOr14b that together determine the functional shift to its ortholog, HarmOr14b, in *H. armigera*. The peripheral modifications of the two closely related species took place in both PR expression level and PR tuning selectivity. These findings not only help us specifically understand the evolution of the two *Helicoverpa* species, but also provide new insights into the structure and function of cell membrane receptors.

## Materials and methods

### Insects

*H. assulta* and *H. armigera* were originally collected as larvae in tobacco fields in Zhengzhou, Henan province of China, and were reared at the Institute of Zoology, Chinese Academy of Sciences, Beijing. The larvae were fed with an artificial diet, mainly composed of wheat germ, yeast and chili for *H. assulta*, wheat germ, yeast and tomato paste for *H. armigera*. Rearing took place at a temperature of 26 ± 1°C with a photoperiod of 16L:8D and 55–65% relative humidity. Male and female pupae were placed in separate cages for eclosion. A 10% honey solution was used as the diet for adults. Virgin adults at 1–3 days old were used in all experiments.

### *Xenopus laevis*

All procedures were approved by the Animal Care and Use Committee of the Institute of Zoology, Chinese Academy of Sciences for the care and use of laboratory animals. Female *X. laevis* were provided by Prof. Zhan-Fen Qin from Research Center for Eco-Environmental Sciences, Chinese Academy of Sciences, and reared with pig liver as food in our laboratory. A total of nine healthy naive *X. laevis* with 18–24 months of age were at the time of the experiment. They were group housed in the box with purified water in 20 ± 1°C. The surgery was performed following the reported protocols (*Nakagawa and Touhara, 2013*). *X. laevis* were anesthetized by bathed in the mixture of ice and water in 30 min, and the oocytes were surgically collected before experiments.

### Sequencing and PR genes of *H. assulta* expression analysis

Total RNA was extracted using the TRIzol reagent (Invitrogen, Carlsbad, CA) and treated with RNase-free DNase I. Poly(A) mRNA was isolated using oligo dT beads. First-strand complementary DNA was generated using random hexamer-primed reverse transcription, followed by synthesis of the second-strand cDNA using RNaseH and DNA polymerase I. Paired-end RNA-seq libraries were prepared following Illumina's protocols and sequenced on the Illumina HiSeq 2000 platform (San Diego, CA). The RNA-seq reads were mapped using Bowtie2 (*Langmead and Salzberg, 2012*). Gene expression levels were measured using the reads per kb per million mapped reads criterion (FPKM). FPKM values were calculated by custom python script (https://github.com/ningchaozky/fpkm-calculate-from-bam-or-sam-.git [*Ning, 2017*]; copy archived at https://github.com/elifesciences-publications/fpkm-calculate-from-bam-or-sam-.git)*Ning, 2017*. Only genes with a FPKM >1 and coverage more than 0.6-fold of transcripts were used for further analysis. Differentially expressed genes were detected using the DEGseq (RRID: SCR_008480) (*Wang et al., 2010*), which was constructed based on the Poisson distribution and eliminated the influences of sequencing depth and gene length. Annotation of PR genes was performed by NCBI blastx against a pooled insect PR database and then the expression was extracted from the DEGseq result.

### Phylogenetic analysis

Phylogenetic analysis of PRs was performed based on amino acid sequences contained in reports of PRs of Noctuidae. The phylogenetic tree was constructed using the MEGA6.0 program (RRID: SCR_

000667) with neighbor-joining phylogeny using the p-distances model (*Tamura et al., 2013*). Node support was assessed using a bootstrap procedure based on 1000 replicates. The ratios of nonsynonymous to synonymous substitutions (dN/dS) were computed using DnaSP version 5.10 (RRID: SCR_003067) (*Librado and Rozas, 2009*).

## RNA isolation and cDNA synthesis

The antennae from three-day-old virgin adults were dissected and immediately collected into a 1.5 mL Eppendorf tube, containing liquid nitrogen, and stored at −80°C until use. Total RNA was extracted by QIAzol Lysis Reagent following the manufacturer's protocol (including DNase I treatment). RNA quality was checked with a spectrophotometer (NanoDrop 2000, Wilmington, DE). The single-stranded cDNA templates were synthesized using 2 μg total RNAs from various samples with 0.5 μg oligo (dT) 15 primer (Promega, Madison, WI), heated to 70°C for 5 min to melt the secondary structure within the template, then using M-MLV reverse transcriptase (Promega) at 42°C for 1 hr, and stored at −20°C.

## Quantitative real-time PCR

qPCR was performed on an Mx3005P qPCR System (Agilent Technologies, Palo Alto, CA) with SYBR Premix Ex *Taq* (TaKaRa, Shiga, Japan). The gene-specific primers to amplify an 80–150 bp product were designed by Primer-BLAST (http://www.ncbi.nlm.nih.gov/tools/primer-blast/), and are listed in *Supplementary file 1*. The qPCR reaction was: 10 s at 95°C, followed by 40 cycles of 95°C for 5 s and 60°C for 31 s, followed by the measurement of fluorescence during a 55°C to 95°C melting curve to detect a single gene-specific peak, and to check the absence of primer dimer peaks. The product was verified by nucleotide sequencing. 18S ribosomal RNA (GenBank number: EU057177.1) was used as the control gene. Each reaction was run in triplicate (technical replicates) and the means and standard errors were obtained from three independent biological replicates. The relative copy numbers of PR genes were calculated according to the $2^{-\Delta\Delta Ct}$ method (*Livak and Schmittgen, 2001*).

## Cloning of the candidate pheromone receptor of *H. assulta* and *H. armigera*

Based on the full-length nucleotide sequences of PRs in *H. assulta* or *H. armigera* (GenBank numbers are listed in *Supplementary file 2*), specific primers were designed and are reported in *Supplementary file 1*. All amplification reactions were performed using Q5 High-Fidelity DNA Polymerase (New England Biolabs). The PCR conditions for the PRs were: 98°C for 30 s, followed by 30 cycles of 98°C for 10 s, 50°C for 30 s and 72°C for 1 min, and extension at 72°C for 2 min. Templates were obtained from male or female antennae of *H. assulta* or *H. armigera*. The sequences were verified by both the Sanger sequencing for 10 samples, and the transcriptome data.

## *In situ* hybridization

Two-color double *in situ* hybridizations were performed following protocols reported previously (*Krieger et al., 2002*; *Ning et al., 2016*). The sense and antisense primers were used to synthesize the gene-specific probes from the open-reading frames (*Supplementary file 1*). Both digoxin (Dig)-labeled and biotin (Bio)-labeled probes were synthesized by DIG RNA labeling Kit version 12 (SP6/T7) (Roche, Mannheim, Germany), with Dig-NTP or Bio-NTP (Roche, Mannheim, Germany) labeling mixture, respectively. RNA probes were subsequently fragmented to 300 nt by incubation in carbonate buffer. Antennae were dissected from 2- to 4-day-old male moths, embedded in JUNG tissue freezing medium (Leica, Nussloch, Germany) and frozen at −80°C until use. Sections (12 μm) were prepared with a Leica CM1950 microtome at −22°C, then mounted on SuperFrost Plus slides (Thermo Scientific, Waltham, MA). After a series of fixing and washing procedures, 100 μL hybridization solution (Boster, Wuhan, China) containing both Dig and Bio probes was placed onto the tissue sections. A coverslip was added and slides were incubated in a humid box at 55°C overnight. After hybridization, slides were washed twice for 30 min in 0.1 × saline sodium citrate (SSC) at 60°C, treated with 1% blocking reagent (Roche, Mannheim, Germany) in TBST (100 mM Tris, pH = 7.5, 150 mM NaCl with 0.03% Triton X-100) for 30 min at room temperature, and then incubated for 60 min with anti-digoxigen (Roche, Mannheim, Germany) and Strepavidin-HRP (PerkinElmer, Boston, MA).

Hybridization signals were visualized by incubating the sections for 30 min with HNPP/Fast Red (Roche, Mannheim, Germany), followed by three 5 min washes in TBS with 0.05% Tween-20 (Tianma, Beijing, China) at room temperature, with shaking. The sections were incubated with Biotinyl Tyramide Working Solution for 8 min at room temperature followed by the tyramide-signal amplification (TSA) kit protocols (PerkinElmer, Boston, MA). After three additional washings for 5 min in TBS with 0.05% Tween-20 at room temperature with shaking, sections were finally mounted in Antifade Mounting Medium (Beyotime, Beijing, China). All the sections were analyzed under a Zeiss LSM710 Meta laser scanning microscope (Zeiss, Oberkochen, Germany). Adobe Illustrator CS6 (RRID: SCR_014198) (Adobe systems, San Jose, CA) was used to arrange figures only to adjust brightness and contrast.

## Construction of the mutation sequences

To generate the regional mutations, we first cloned each fragment of the sequences. The mutation fragment was cloned using the primer of Mutant-F and Mutant-R, with the cDNA of *H. armigera*. The other parts were cloned using the primer of HassOr14b-F/HassOr14b-Fragment1-R, which generated the first fragment, and HassOr14b-Fragment2-F/HassOr14b-R, which generated the second fragment, with the cDNA of *H. assulta*. The mutation fragment had 25–60 bp overlap sequences with the other two fragments. The conditions were: 98°C for 30 s, followed by 25 cycles of 98°C for 10 s, 52°C for 30 s and 72°C for 30 s, and extension at 72°C for 2 min. Then we used the primers of HassOr14b-F/HassOr14b-R, with the mixture of purified fragment products as the template, to generate the regional mutation sequences. The conditions were: 98°C for 30 s, followed by 20 cycles of 98°C for 10 s, 52°C for 30 s and 72°C for 90 s, and extension at 72°C for 2 min. The construction diagram was presented in *Figure 7—figure supplement 1*. For the site mutations, we used the primer of HassOr14b-F/mutation1 R to generate the first fragment, and the mutation2-F/HassOr14 b-R to generate the second fragment. Then we used the primer of HassOr14b-F/HassOr14b-R, with the mixture of purified fragment products as the template, to generate the site mutation sequences. The conditions were the same as for the construction of regional mutation sequences. The primers are listed in *Supplementary file 1*.

## Receptor functional analysis

The full-length coding sequences of PRs and mutations were first cloned into pGEM-T vector (Promega) and then subcloned into pCS2$^+$ vector. cRNAs were synthesized from linearized modified pCS2$^+$ vectors with mMESSAGE mMACHINE SP6 (Ambion, Austin, TX). Mature healthy oocytes were treated with 2 mg mL$^{-1}$ of collagenase type I (Sigma-Aldrich, St. Louis, MO) in Ca$^{2+}$-free saline solution (82.5 mM NaCl, 2 mM KCl, 1 mM MgCl$_2$, 5 mM HEPES, pH = 7.5) for 20 min at room temperature. Oocytes were later microinjected with 27.6 ng PR cRNA and 27.6 ng Orco cRNA. Distilled water was microinjected into oocytes as a negative control. Injected oocytes were incubated for 3–5 days at 16°C in bath solution (96 mM NaCl, 2 mM KCl, 1 mM MgCl$_2$, 1.8 mM CaCl$_2$, 5 mM HEPES, pH = 7.5) supplemented with 100 mg mL$^{-1}$ gentamycin and 550 mg mL$^{-1}$ sodium pyruvate. Whole-cell currents were recorded with a two-electrode voltage clamp. Intracellular glass electrodes were filled with 3 M KCl and had resistances of 0.2–2.0 MΩ. Signals were amplified with an OC-725C amplifier (Warner Instruments, Hamden, CT) at a holding potential of −80 mV, low-pass filtered at 50 Hz and digitized at 1 kHz. Data acquisition and analysis were carried out with Digidata 1322A and pCLAMP software (RRID: SCR_011323) (Axon Instruments Inc., Foster City, CA). Dose-response data were analyzed using GraphPad Prism (RRID: SCR_002798 6) (GraphPad Software Inc., San Diego, CA).

## Data analysis

Response values are indicated as mean ± SEM. Data were square-root transformed and differences were considered significant when $p < 0.05$. n represents number of sections in all cases. One-way ANOVA and Tukey HSD tests with two distribution tails were performed using the Statistical Program for Social Sciences 22.0 (RRID: SCR_002865) (IBM Inc., Armonk, NY).

## Acknowledgements

We thank our colleagues Hao Guo, Lin Yang, Ya-Lan Sun, Rui Tang for their kind assistances in *in situ* hybridization, confocal microscopy, the construction of mutation sequences, and data analysis, respectively. We thank Dr. Ya-Nan Zhang from Huaibei Normal University and Dr. Da-Song Chen from Guangdong Institute of Applied Biological Resources for their kind assistances in phylogenetic analysis. We thank Prof. Zhan-Fen Qin from Research Center for Eco-Environmental Sciences, Chinese Academy of Sciences for providing *Xenopus laevis* frogs. We thank Prof. Paolo Pelosi from University of Pisa, Italy and Prof. Bill Hansson from Max Planck Institute for Chemical Ecology, Germany for valuable comments. This work is supported by the Strategic Priority Research Program of the Chinese Academy of Sciences (grant number XDB11010300), the National Natural Science Foundation of China (grant number 31130050), the National Key R and D Program of China (grant number 2017YFD0200400), and the National Basic Research Program of China (grant number 2013CB127600).

## Additional information

### Funding

| Funder | Grant reference number | Author |
| --- | --- | --- |
| Strategic Priority Research Program of the Chinese Academy of Sciences | Grant number XDB11010300 | Chen-Zhu Wang |
| National Natural Science Foundation of China | Grant number 31130050 | Chen-Zhu Wang |
| National Basic Research Program of China | Grant number 2013CB127600 | Chen-Zhu Wang |
| National Key R&D Program of China | Grantnumber 2017YFD0200400 | Chen-Zhu Wang |

The funders had no role in study design, data collection and interpretation, or the decision to submit the work for publication.

### Author contributions

Ke Yang, Data curation, Software, Formal analysis, Validation, Investigation, Visualization, Methodology, Writing—original draft, Writing—review and editing, Experiment design; Ling-Qiao Huang, Resources, Funding acquisition, Validation, Writing—original draft, Writing—review and editing; Chao Ning, Formal analysis, Validation; Chen-Zhu Wang, Conceptualization, Resources, Data curation, Software, Formal analysis, Supervision, Funding acquisition, Validation, Investigation, Visualization, Methodology, Writing—original draft, Project administration, Writing—review and editing, Experiment design

### Author ORCIDs

Ke Yang http://orcid.org/0000-0002-4138-3373
Chen-Zhu Wang http://orcid.org/0000-0003-0418-8621

### Ethics

Animal experimentation: All procedures in this study were approved by the Animal Care and Use Committee of the Institute of Zoology, Chinese Academy of Sciences for the care and use of laboratory animals (protocol number IOZ17090-A). The surgery was performed following the protocols reported by Nakagawa and Touhara (2013). The *Xenopus laevis* was anesthetized by bathed in the mixture of ice and water in 30 min, the wounds were carefully treated to avoid infection. Every effort was made to minimize suffering.

### Decision letter and Author response

Decision letter https://doi.org/10.7554/eLife.29100.022

Author response https://doi.org/10.7554/eLife.29100.023

## Additional files

### Supplementary files

• Supplementary file 1. Primers used for qPCR, *in situ* hybridization (*In situ*), *Xenopus* oocytes expression (XE) and amino acid mutation (MUT).
DOI: https://doi.org/10.7554/eLife.29100.019

• Supplementary file 2. The accession numbers of all PRs and Orcos used in phylogenetic analysis.
DOI: https://doi.org/10.7554/eLife.29100.020

• Transparent reporting form
DOI: https://doi.org/10.7554/eLife.29100.021

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
