## [Decision Letter]

[Editors’ note: the authors were asked to provide a plan for revisions before the editors issued a final decision. What follows is the editors’ letter requesting such plan.]

Thank you for submitting your article "Two single-point mutations shift the ligand selectivity of a pheromone receptor between two sister moth species" for consideration by *eLife*. Your article has been reviewed by Fred Gould (Reviewer #1), Astrid Groot (Reviewer #2), and Christer Loefstedt (Reviewer #3), and the evaluation has been overseen by a Reviewing Editor and Ian Baldwin as the Senior Editor.

The reviewers have discussed the reviews with one another and the Reviewing Editor has drafted this letter to prompt a response from you concerning the serious concerns of the reviewers. Please address these concerns in a letter, which we will have the Board and reviewers consider before a binding recommendation is made.

Your manuscript has been evaluated by three experts in the field. Although they found your results interesting, they have also identified important concerns related to the experimental results in connection to previous studies by your and other groups. As a consequence, the story that you tell in the manuscript is flawed. For the manuscript to be acceptable for publication in *eLife*, you need to connect better to the literature and include caveats such as the suitability of the *Xenopus* system that does not seem to work well for HR14b in Hass and the fact that there are other receptors for Z9-16Ald in Hass and that Z9-16Ald may be antagonistic in both species. How does this affect the conclusions that you can draw from your experiments? Also the in situ hybridization images are not convincing and do not support previous publications. You will find the detailed comments by the reviewers below. In the light of these evaluations, the manuscript cannot be accepted at this moment. If you feel that you can effectively address the most critical concerns of the reviewers, please send us your responses for further evaluation but the Board and reviewers.

*Reviewer #1:*

General: Overall this is a very interesting paper that is one of only a couple of papers that have identified amino acid changes if a moth pheromone receptor that change specificity. This is a very important finding because it could show that the evolution of new sexual communication systems in moths could evolve based on very simple genetic changes. That said, I do have some concerns about specific issues.

Specific:

1) As far as I can tell only one peer reviewed paper indicates the Z9-14Ald is an active positive part of the Harm pheromone. In another paper this compound is found to be an antagonist in both Hass and Harm. This should be made clear to the reader because unlike some other systems where there is a clear change from using one compound to using another in two closely related species, in this case the selective pressure to evolve the receptor in Harm for Z9-14Ald is not very clear.

2) Earlier research papers that have searched for activity of HR14b in Hass have failed to find any activity. This includes a recent paper by some of the authors of the current paper. It would be useful for the authors to explain why they did not find activity in their other recent paper, but that they now find it.

3) Looking at the plots in the figures that show "current (nA)" on the Y-axis, it becomes clear that at least with the *Xenopus* system, the level of activity of the Hass HR14b receptor is much lower than for the Harm HR14b receptor. Whereas the response to Z9-14Ald in Harm is over 400 and for Z9-16Ald is around 50, the response of the Hass HR14b receptor to even Z9-16Ald (to which it is specific) is only about 60. Clearly, this oocyte system is not very efficient for Hass HR14b. This may be why previous studies didn't find any activity. The authors should acknowledge this.

4) This lack of sensitivity of the HR14b of Hass is problematic for interpreting the results of single mutations to the Hass HR14b receptor. Clearly, the T3551 mutation dramatically increases the overall response of HR14b while negating specificity. This makes it hard to interpret the interaction of the two mutations.

5) There is overlap between the paper Olfactory perception and behavioral effects of sex pheromone gland components in *Helicoverpa armigera* and *Helicoverpa assulta*, Meng Xu et al. 2016 and the current paper. This overlap should be made more clear.

*Reviewer #2:*

This manuscript describes the functional characterization of an olfactory receptor in the noctuid moth *Helicoverpa assulta*, HassOR14b, that is tuned to the major sex pheromone of this species, Z9-16:Ald, while its ortholog HarmOR14b in *H. armigera* is tuned to Z9-14:Ald, the secondary sex pheromone component of this species (the main sex pheromone component Z11-16:Ald, which is perceived through HarmOR13). Through a series of mutagenesis experiments, the authors convincingly show that two single point mutations, F232I and T355I, in HassOR14b changes the sensitivity of this receptor from Z9-16:Ald to Z9-14:Ald. These mutations correspond to the HarmOR14b amino acid sequences at these sites. In additions, the authors show with in-situ hybridizations that HassOR14b is co-localized with HassOR6 and HassOR16.

I only have a few minor, but essential comments on the text that I think can easily be addressed:

1. I miss citation to the recent article by De Fouchier et al. in Nature Communications (DOI: 10.1038/ncomms1570910)

2. Since HarmOR14b is tuned to Z9-14:Ald, I think it's important that the authors write 1-2 sentences on how this component is a minor (but essential) sex pheromone component in *H. armigera*. In the current text it seems that the sex pheromone system of these two species is a two-component system with similar reverse ratios as in the two pheromone strains of Ostrinia nubilalis, while the pheromone blend of H. armigera (and also H. assulta) is a bit more complex than just a two-component blend.

3. The authors give dN/dS ratios (ω) for the different ORs (Results, subsection “Phylogenetic analysis of candidate PRs”), reasoning that ω < 1 indicates puryfying selection, while ω > 1 indicates positive selection. As they found a ω = 0.17 for cluster OR14b, this thus indicates puryfying selection. However, in the discussion the authors do not come back to this result and instead write "The female moth produces a pheromone blend of several components, stabilized by strong selection pressure against any change in such blends (Roelofs et al., 2002). This requires an equivalent stability from the male moths to detect the same species-specific pheromones, but at the same time should allow for a degree of plasticity to adapt to changes in pheromone structures associated with speciation." This part needs to be revised, as 'stabilized by strong selection pressure' and 'equivalent stability' are strangely used. Also, the latter part "should allow for a degree of plasticity" comes across as hand waving. Similarly, "In the course of speciation, the functional change of ORs is a gradual process with multiple amino acid mutations, a few making drastic changes and many making small modifications or even no change in function" need to be revised, as 'gradual changes' contradicts 'a few making drastic changes' and as a whole this sentence doesn't make sense.

4. Discussion, subsection “Novel identified PR and the different combinations with other PRs”, first paragraph: HassOR14b should be "HassOR16 in the same sensilla". Do the authors know where the orthologous ORs are localized in *H. armigera*? Is HarmOR14b also co-localized with HarmOR6 or HarmOR16? Can I deduce from the intro information that these sensilla are the type C sensilla?

5. Discussion, subsection “Novel identified PR and the different combinations with other PRs”, second paragraph:: Where are these amino acid residues located? I think it's important to specify this, especially because this is the first study (right?) where amino acid sequence changes in the intracellular domains (ICDs). I would also like to read a bit more on how the authors think these changes may alter the function. They give two very short explanations in the next paragraph, but what do they mean with 'complex deep structure'?

*Reviewer #3:*

Yang et al. report that two amino acid substitutions in intracellular domains may account for the difference in ligand specificity between HarmOr14b and HassOr14b. The conclusion is based on a series of mutagenesis experiments. The results are interesting but "the story" is not ready for publication. A number of major issues are listed below. In addition, I think that the discussion of how mutations in the intracellular domains may influence ligand specificity (subsection “Two amino acids located in the intracellular domains together determine the OR selectivity”, last paragraph) is speculative and lacks both references and a possible mechanism.

The study is mainly based on the authors´ finding that HassOR14b is responsive to Z9-16:Ald whereas HarmOR14b is responsive to Z9-14:Ald. However, according to a previous study from the same laboratory (Jiang et al., 2014) and the study by Chang et al. (2015), HassOR14b did not respond to any tested compounds including Z9-16:Ald. This needs to be clarified. The authors do not mention about this discrepancy and do not even cite the Chang et al. paper:

Jiang, X.-J., Guo, H., Di, C., Yu, S., Zhu, L., Huang, L.-Q., Wang, C.-Z. (2014). Sequence similarity and functional comparisons of pheromone receptor orthologs in two closely related *Helicoverpa* species. Insect Biochemistry and Molecular Biology 48, 63-74.

Chang, H., Guo, M., Wang, B., Liu, Y., Dong, S., and Wang, G. (2016). Sensillar expression and responses of olfactory receptors reveal different peripheral coding in two *Helicoverpa* species using the same pheromone components. Scientific reports, 6, 18742.

In the phylogenetic tree, the authors only included Heliothine species in OR13 cluster, but not the orthologues from other species, thus the calculation of dN/dS value is biased. Moreover, the authors say the dN/dS value of OR11 cluster is larger than 1.8, this is inconsistent with the previous findings that dN/dS value of this cluster is low (around 0.1):

Zhang, D. D., and Löfstedt, C. (2013). Functional evolution of a multigene family: orthologous and paralogous pheromone receptor genes in the turnip moth, Agrotis segetum. PLoS One, 8(10), e77345.

Zhang, Y. N., Zhang, J., Yan, S. W., Chang, H. T., Liu, Y., Wang, G. R., and Dong, S. L. (2014). Functional characterization of sex pheromone receptors in the purple stem borer, Sesamia inferens (Walker). Insect molecular biology, 23(5), 611-620.

The in situ hybridization images are not convincing and this is problematic as the results do not correspond to what was reported in Chang et al. (2015). These authors claimed that HassOR6 and HassOR16 are localise in the same sensillum (Chang et al., 2015).

Discussion subsection “Novel identified PR and the different combinations with other PRs”, first paragraph: 'This suggests that the ancestor of the two sister species not only changed OR14b's expressing level, but also altered its tuning selectivity to meet the species specific demands': The authors did not compare the expression levels of HassOR14b and HarmOR14b so the statement is not supported. In addition, it is not clear to me how the authors can conclude anything about the expression levels in the ancestor. Based on analysis of the contemporary species (which at some point had a common ancestor) we can at the best conclude that expression levels are different in these two species. The reasoning further involves a teleological argument, i.e. that the species that evolved from the common ancestor had some "specific demands". This is not how evolution works.

In the subsection “Quantitative real-time PCR”: the authors write that the reference gene is 18s rRNA, but the GenBank number provided is actually the actin gene. This is confusing.

In the subsection “Construction of the mutation sequences”: The description of the construction strategy would benefit from a diagram visualizing the different steps.

[Editors’ note: formal revisions were requested, following approval of the authors’ plan. After the authors submitted their revised paper, further revisions were requested prior to acceptance, as described below]

Thank you for submitting your article "Two single-point mutations shift the ligand selectivity of a pheromone receptor between two closely related moth species" for consideration by *eLife*. Your article has been reviewed by Fred Gould (Reviewer #1), Astrid Groot (Reviewer #2), and Christer Loefstedt (Reviewer #3), and the evaluation has been overseen by a Reviewing Editor and Ian Baldwin as the Senior Editor.

The reviewers have discussed the reviews with one another and the Reviewing Editor has drafted this decision to help you prepare a revised submission.

The reviewers have appreciated your effective revision of the manuscript and the extensive explanation in the rebuttal. They conclude that your study provides very interesting results on how small differences in pheromone receptors may influence ligand selectivity. Still some important issues remain and I invite you to address these comments by the reviewers and prepare a second revision of the manuscript. Specific attention should be paid to sentences where you refer to the literature on odour reception and genetic mechanisms and evolutionary consequences. Some of these sentences are highlighted in the reviews, while others are not. For example, what is meant with 'a few steps' in the last sentence of the Abstract?

*Reviewer #1:*

The *eLife* editor summarized my concerns well as "For the manuscript to be acceptable for publication in *eLife*, you need to connect better to the literature and include caveats such as the suitability of the *Xenopus* system that does not seem to work well for HR14b in Hass and the fact that there are other receptors for Z9-16Ald in Hass and that Z9-16Ald may be antagonistic in both species. How does this affect the conclusions that you can draw from your experiments? "

The authors have made useful changes but I think a little more discussion of some of the issues with the *Xenopus* system in this specific case is warranted because the same issue is likely to arise in future studies.

I realize that in the current Abstract and in most of the manuscript the authors avoid discussing the relevance of their findings to the evolution of the two species of *Helicoverpa*. For example, in the Abstract they state that "We conclude that species-specific changes in the tuning specificity of the PRs of male moths could be achieved with just a few steps". As long as the authors don't indicate that their findings help us to specifically understand the evolution of the two *Helicoverpa* moths, I think they are on solid ground. The two mutations they study certainly change the tuning specificity.

It still would be good if the authors could elaborate on the fact their findings are of most interest in terms of neurophysiology and are not a direct commentary on the evolution of these specific moths.

*Reviewer #2:*

Overall, I'm very impressed with all the work the authors have done and also with their revision. I have a few remaining questions for the authors:

The numbering of the figures was a bit confusing, I'm not sure which figures are now supposed to be supplementary figures, so I'll go with the complete names of the figures.

1) In Figure 3 the authors show that HasOR14b responds to Z9-16:Ald, but when comparing the different graphs, the y-axis in Figure 3 is in a different (smaller) scale than the y-axis in Figure 3—figure supplement 1, which shows that HassOr6 actually responds more to Z9-16:Ald (300 nA) than HassOr14b (70 nA). Such a difference in scaling is also present in Figure 8; HassOr14b shows a current response to Z9-16:Ald of 70 nA again, and/but HarmOr14b shows a current response of 30-40 (?) nA. I do understand that HarmOr14b mostly responds to Z9-14:Ald, but the response to Z9-16:Ald may also be important, no? (In this respect it is indeed very impressive that the authors found an almost 10x stronger response when mutating T335I (see Figure 7—figure supplement 2). Which relates to my next question:

2) As Z9-16:Ald is also an important (albeit minor) sex pheromone component of *H. armigera*, with what receptor(s) do the authors think *H. armigera* is perceiving this component?

3) In Figure 2 the authors show that HassOR6 is highly expressed in males, and a little in females. Which makes sense to me, as females probably smell their own pheromone as well. I don't think that sex-specific expression is a necessary criterium for an OR to be a sex pheromone OR.

4) The authors have responded to previous comments on why previous functional assays with HassOr14b didn't work by writing that the sequence had not been correct in previous work and that this is now corrected. In Figure 3—figure supplement 3 they specify which 3 amino acids have been corrected (Yang et al. is the current sequence used). In checking these 3 amino acids in Figure 6 found them in RIII, in RVII and in RVIII. I find it hard to determine whether one of these 3 SNPs were among the ones tested in Figure 7—figure supplement 2. Still, I think the authors make a convincing case that the right sequence together with the selection of the appropriate expression vector is crucial to identify the function of receptors.

5) Hopefully Figure 7—figure supplement 2 is not a supplementary figure, as I think this figure is a great way of showing the results of the single mutations.

6) As for the in situ hybridization pictures, these are clear to me, but I leave this to the more experienced reviewer(s) to decide whether this is sufficiently clear.

*Reviewer #3:*

In summary, I think that the revision represents a considerable improvement of the manuscript and the authors have to a large extent explained many of the mistakes and the missing information in the original submission. However, some of my previous comments/questions still apply.

In the rebuttal letter the authors write:

'Basically there are two major concerns: (Baker et al., 2004) the suitability of the *Xenopus* system […] (Boo et al., 1995) Contradiction of our in situ hybridization results'.

In my opinion this is not correct. At least not when it comes to my concerns. My major concern was that the authors avoided mentioning the inconsistency in the response of wild type HassOR14b compared to previous publications. In the current version, the authors explain that the gene sequences were not correct in previous publications. However, the explanation is not straight forward:

- The sequences in Chang et al., 2016 were based on the transcriptome data from the previous paper (Zhang et al., 2015) from the same group. There's a single amino acid difference of HassOR14b sequence between the current manuscript and the sequence in Chang's paper. Could it simply be explained by the use of different polymerases? How could the authors exclude the possibility of polymorphism? Although possible, it's not very likely that this one amino acid difference would totally abolish the response to Z9-16:Ald (No response to Z9-16:Ald or any other ligand in Chang et al. 2016) but significant response to Z9-16:Ald in the current study (70 nA – Figure 8). The authors of the present manuscript can of course not take the responsibility of any possible flaws in a study published by another group but the discrepancy requires an explicit comment in the Discussion.

Zhang, J., Wang, B., Dong, S., Cao, D., Dong, J., Walker, W. B.,.… & Wang, G. (2015). Antennal transcriptome analysis and comparison of chemosensory gene families in two closely related noctuidae moths, *Helicoverpa armigera* and *H. assulta*. PloS one, 10(2), e0117054.

I do not understand how different expression vectors could cause the difference in sequence. The authors have to explain how. Otherwise it is just an ad hoc explanation that needs to be tested. It's the cRNAs that are injected and expressed in the oocytes. The resulting receptor proteins should have the same sequences despite of what the original expression vectors were. In addition, the pT7TS vectors have been widely and successfully used in oocyte recordings and OR studies. Why should this vector not be suitable for HassOR14b expression and oocyte recordings?

The authors write in the rebuttal and mention in the Discussion that they re-sequenced HassOr14b and came up with the "right sequence". This is a new (and important) result and as such it should be mentioned in the Results section and the relevant methods and materials should be described in Materials and methods before the discrepancy with previous studies is discussed in the Discussion section.

As far as I understand reviewer #1 was not really questioning oocytes as a suitable system for OR de-orphanization in general, but said that the system may not be efficient for testing HassOR14b. In my opinion, however, a current of 60 nA is not too small to be noticed. Hence this is not the reason why Chang et al. did not find a response, but there seems to be a real difference in activity between the two studies.

It remains mysterious to me why the wild type HassOR14b responds so differently in this and previous studies and it's problematic that the authors did not mention this inconsistency in the first version. In addition, a number of other mistakes/errors were pointed out by me and the other reviewers. The authors have now corrected these errors but can we be sure that additional errors did not go unnoticed by the reviewers? Careful double checking appears necessary.

Some other points for your consideration.

1) 'Unlike general odorant receptors (ORs) that typically bind more than one ligand (de Fouchier et al., 2017), PRs are narrowly tuned to specific pheromone components (Grosse-Wilde et al., 2007).' This statement is not correct. There are both narrowly tuned PRs and broadly tuned PRs reported in the literature (Miura et al., 2010; Wanner et al. 2010; Zhang and Löfstedt 2015). Even in Grosse-Wilde et al., 2007 cited by the authors, it was proposed that 'there are two different designs of pheromone receptors'. The authors have to be more precise in their statements and more rigorous when citing the literature:

Miura, N., Nakagawa, T., Touhara, K., and Ishikawa, Y. (2010). Broadly and narrowly tuned odorant receptors are involved in female sex pheromone reception in Ostrinia moths. Insect biochemistry and molecular biology, 40(1), 64-73.

Wanner, K. W., Nichols, A. S., Allen, J. E., Bunger, P. L., Garczynski, S. F., Linn Jr, C. E., […] and Luetje, C. W. (2010). Sex pheromone receptor specificity in the European corn borer moth, Ostrinia nubilalis. PLoS One, 5(1), e8685.

Zhang, D. D., and Löfstedt, C. (2015). Moth pheromone receptors: gene sequences, function, and evolution. Frontiers in Ecology and Evolution, 3, 105.

2) 'For example, BmOr1 of.[…] to 1,3Z,6Z,9Z-19:H (Zhang et al., 2016).' As we suggested before, there's no point of listing these examples so this sentence can be removed.

3) Results and Discussion sections: ‘all the PRs and Orcos are subjected to purifying selection which is consistent with the previous studies (Zhang and Löfstedt, 2013; Zhang et al., 2014b)'. This is not a correct reference. Zhang and Löfstedt (2013) as well as other researchers (e.g. Leary et al., 2012) found that some PR clusters are under purifying selection while some are under relatively relaxed selective pressure. Since the authors used a limited number of moth species in the phylogenetic tree in the current study, it's too daring to draw the conclusion that 'all the PRs and Orcos are subjected to purifying selection'.

Leary, G. P., Allen, J. E., Bunger, P. L., Luginbill, J. B., Linn, C. E., Macallister, I. E., […] and Wanner, K. W. (2012). Single mutation to a sex pheromone receptor provides adaptive specificity between closely related moth species. Proceedings of the National Academy of Sciences, 109(35), 14081-14086.

4) In situ results: 'in some sensilla only HassOr14b was detected'. For the Discussion: Is this consistent with previous SSR data? Which gene is supposed to be expressed in the neighboring neuron?

5) The authors refer to the transcriptome a couple of times in the manuscript. Have the authors deposited the transcriptome data in a public database?

6) Discussion subsection “Implications for the modulation and evolution of OR selectivity”: 'The olfactory systems found in all animals have nearly the same design feathers, which give olfaction a considerably flexibility for signaling to evolve.' I don't understand this sentence. What does 'design feathers' mean? Should it be "features"?

7) The English still needs some attention throughout the manuscript before its scientific merits can be definitely evaluated.

---

## [Author Response]

[Editors’ note: what follows is the authors’ plan to address the revisions.]

Your manuscript has been evaluated by three experts in the field. Although they found your results interesting, they have also identified important concerns related to the experimental results in connection to previous studies by your and other groups. As a consequence, the story that you tell in the manuscript is flawed. For the manuscript to be acceptable for publication in eLife, you need to connect better to the literature and include caveats such as the suitability of the Xenopus system that does not seem to work well for HR14b in Hass and the fact that there are other receptors for Z9-16Ald in Hass and that Z9-16Ald may be antagonistic in both species. How does this affect the conclusions that you can draw from your experiments? Also the in situ hybridization images are not convincing and do not support previous publications. You will find the detailed comments by the reviewers below. In the light of these evaluations, the manuscript cannot be accepted at this moment. If you feel that you can effectively address the most critical concerns of the reviewers, please send us your responses for further evaluation but the Board and reviewers.

Basically there are two major concerns: (Baker et al., 2004) the suitability of the *Xenopus* system for functional analysis of pheromone receptors like HassOr14b; (Boo et al., 1995) Contradiction of our in situhybridization results with Chang et al. (2016) in Scientific Reports. “Z9-16:Ald” you mentioned above could be “Z9-14:Ald”?

For the first major concern, the *Xenopus laevis* oocyte expression system is a valuable tool for the study of function of insect ORs, up to now, the function of many insect ORs have been characterized successfully. We think this system is also suitable for HassOr14b based on the following reasons:

1) The responding current to 10^-4^ M of Z9-16:Ald is about 70 nA, which is comparable to the current (80 nA) of the oocyte expressing HarmOr13 tuning to the major component, Z11-16:Ald in *H. armigera*. We have done many technical and biological replications, and the results are repeatable.

2) The oocytes expressing HassOr14b to Z9-16:Ald are retested within a dose–response paradigm, revealing a clear concentration dependency.

3) Other evidences also support the HassOr14b is the receptor of Z9-16:Ald in *H. assulta*: The expression level of HassOr14b is the highest among all the PRs in male antennae of *H. assulta*, and the OSNs responding to Z9-16:Ald is most abundant in male antennae of *H. assulta*; the localization patterns of HassOr14b with other PRs in male antennae of *H. assulta* are consistent with the single sensilla recording results.

Please also find a more detailed response to this question raised by the reviewer #1 later.

For the second major concern, especially raised by the reviewer #3, we provide three additional in situ hybridization images from our experiments to support our point that HassOr14b is co-localized with HassOr6 or HassOr16, and HassOR6 and HassOr16 are not co-localized.

We also carefully examined the related in situ hybridization images (Figure 8) in Chang et al. (2016) (Scientific Reports 6, 18742), which indicated that HassOR6 and HassOr16 are co-localized. From their in situ hybridization images, it is difficult to judge if OR6 and Or16 are co-localized.

For the other questions we answer in the following one by one.

In this study we characterize the function of all the candidate PRs for Z9-16Ald, HassOr14b, HassOr6, HassOr16 in *H. assulta*, and find HassOr14b is the only receptor tuned to Z9-16Ald. We also prove the results of Jiang et al. (2014) that HassOr16 is the receptors in *H. assulta* for Z9-14:Ald, which is an antagonist for this species. For *H. armigera*, Z9-14:Ald acts as an agonist in small amounts (0.3%) but an antagonist in higher amounts (1% and above). We have discussed these previous results, which is helpful to understand the conclusions that we draw from our experiments.

Reviewer #1:[…] 1) As far as I can tell only one peer reviewed paper indicates the Z9-14Ald is an active positive part of the Harm pheromone. In another paper this compound is found to be an antagonist in both Hass and Harm. This should be made clear to the reader because unlike some other systems where there is a clear change from using one compound to using another in two closely related species, in this case the selective pressure to evolve the receptor in Harm for Z9-14Ald is not very clear.

Yes, for *H. armigera*, Z9-14:Ald acts as an agonist in small amounts (0.3%) (Rothschild, 1978; Wu et al., 2015; Zhang et al., 2012) but an antagonist in higher amounts (1% and above) (Gothilf et al., 1978; Kehat and Dunkelblum, 1990; Wu et al., 2015); for *H. assulta,* Z9-14:Ald acts as an antagonist (Boo et al., 1995; Wu et al., 2015). We accept your suggestion and add this information in the revised manuscript (Introduction, fifth paragraph).

The selective pressure to evolve the receptor in *H. armigera* for Z9-14:Ald is intriguing. The SSR data prove that male antennae of *H. armigera* have olfactory sensory neurons (OSNs) responding to Z9-14:Ald (Wu et al. 2015; Xu et al., 2016). Two reasons may explain its evolution:

1) Z9-14:Ald acts as an agonist at the low dosage and an antagonist in the pheromone communication system of *H. armigera*, so it is necessary to have such a pheromone receptor. Up to now, HarmOr14b is the only receptor specifically responding to Z9-14:Ald in *H. armigera.*

2) The pheromone ratio of Z9-16:Ald to Z11-16:Ald in *H. assulta* is 93:7, and that in *H. armigera* is 3:97. Correspondingly, the population ratio of OSNs responding to Z9-16:Ald and Z11-16:Ald in *H. assulta* is 81:19, while that in *H. armigera* is 22:78. Assuming that *H. assulta* is more primitive species than *H. armigera* (Cho et al., 2008; Fang et al., 1997; Wang et al., 2005), shifting a given receptor’s selectivity could be the simplest and most efficient way to change the population ratio of pheromone OSNs in the evolution.

2) Earlier research papers that have searched for activity of HR14b in Hass have failed to find any activity. This includes a recent paper by some of the authors of the current paper. It would be useful for the authors to explain why they did not find activity in their other recent paper, but that they now find it.

It has long been a mystery about the receptor to Z9-16:Ald, the major sex pheromone component of *H. assulta*. In 2014, our group first cloned the full-length sequences of HassOr14b and HarmOr14b by the use of LA-*Taq* polymerase (Takara shuzo, Shiga, Japan), but failed to identify its function when using *Xenopus* system (Jiang et al., 2014). Along with identifying the function of more and more Ors by using *Xenopus* system, we found that the accuracy and integrity of the OR sequence is very important in functional analysis. Therefore, in this study we re-cloned the sequence of *HassOr14b* by use of Q5® High-Fidelity DNA Polymerase (New England Biolabs, USA) and repeated again to verify the sequence by Sanger sequencing for 10 samples, and also compared with the sequence in the antennal transcriptome data of *H. assulta*. Finally, we got the correct sequence, in which there are 3 amino acids different comparing with the old sequences (Figure 3—figure supplement 3). Moreover, we further analyzed the transcriptome data in *H. assulta* and confirmed that in the 3 different amino acids positions, there is no sequence polymorphism. We use the accurate sequence this time and finally get the function of HassOr14b, which is specifically tuned to Z9-16:Ald, the major sex pheromone component in *H. assulta*.

In the study of Chang et al. (2016), they also try to identify the function of HassOr14b but failed when using *Xenopus* system (Chang et al., 2016). There are two differences between the two studies. One is that they also used the LA-*Taq* polymerase we used before when they cloned the sequence, and there is 1 amino acid different in the 5’ ends with ours (Figure 3—figure supplement 3). By analyzing the transcriptome data in *H. assulta,* we confirmed that this amino acid position has no sequence polymorphism. Another difference is the vector used in the expression system. They used the pT7Ts vector, while we use the pCS2+ vector in the expression system.

In short, we find that the accuracy and integrity of the sequence is crucial to identify the function of the receptors. We use the right sequence so we get the activity of HassOr14b. Moreover, using the appropriate expression vector could be also important. We add this information in the second paragraph of the subsection “Novel identified PR and the different combinations with other PRs”.

3) Looking at the plots in the figures that show "current (nA)" on the Y-axis, it becomes clear that at least with the Xenopus system, the level of activity of the Hass HR14b receptor is much lower than for the Harm HR14b receptor. Whereas the response to Z9-14Ald in Harm is over 400 and for Z9-16Ald is around 50, the response of the Hass HR14b receptor to even Z9-16Ald (to which it is specific) is only about 60. Clearly, this oocyte system is not very efficient for Hass HR14b. This may be why previous studies didn't find any activity. The authors should acknowledge this.

The *Xenopus laevis* oocyte expression system with two electrode voltage clamping is an efficient system for identifying insect odorant (pheromone) receptors (Di et al., 2017; Lu et al., 2007; Mitsuno et al., 2008; Nakagawa and Touhara, 2013; Sakurai et al., 2004; Tanaka et al., 2009; Wanner et al., 2007; Xia et al., 2008). One major advantage of oocytes is that these cells only express a few ion channels and receptors, which are quite different from those of insects, so that the insect receptors can be studied without contamination from endogenous channels and receptors (Goldin, 2006). The oocyte system is particularly well suited for the study of many different mutations because injection and two-electrode voltage-clamping can be carried out rapidly and in a semi-automated fashion (Goldin, 2006; Hill et al., 2015; Hughes et al., 2014; Leal, 2013; Leary et al., 2012; Nakagawa et al., 2012).

We think this system is also suitable for HassOr14b based on the following reasons:

1) The responding current of the oocytes expressing HassOr14b to 10^-4^ M of Z9-16:Ald is about 70 nA, which is comparable to the current (80 nA) of the oocyte expressing HarmOr13 tuning to the major component, Z11-16:Ald in *H. armigera* (Liu et al., 2013; our unpublished data). We have done many technical and biological replications, and the results are repeatable.

2) The oocytes expressing HassOr14b to Z9-16:Ald are retested within a dose–response paradigm, revealing a clear concentration dependency.

3) Other evidences also support the HassOr14b is the receptor of Z9-16:Ald in *H. assulta*: The expression level of HassOr14b is highest among all the PRs in male antennae of *H. assulta*, and the OSNs responding to Z9-16:Ald is the most abundant in male antennae of *H. assulta*; the localization patterns of HassOr14b with other PRs in male antennae of *H. assulta* are consistent with the single sensilla recording results.

4) This lack of sensitivity of the HR14b of Hass is problematic for interpreting the results of single mutations to the Hass HR14b receptor. Clearly, the T3551 mutation dramatically increases the overall response of HR14b while negating specificity. This makes it hard to interpret the interaction of the two mutations.

The sensitivity of HassOr14b is well retested by the dose response paradigm of HassOr14b stimulated with a range of Z9-16:Ald concentrations (Figure 4), with the EC50 value for Z9-16:Ald was 8.65 × 10^–5^ M. T355I mutation increases the response of HassOr14b to the related compounds but not overall. We think this does not negate the specificity (not responding to all the tested ligands) but extend the tuning spectrum from Z9-16:Ald to both Z9-16:Ald and Z9-14:Ald.

In the mutagenesis study, the change of the tuning selectivity is what we focus on. We find that only T355I and F232I change the selectivity. When we made the combination of the two site mutations, the PR with two mutations shifted the ligand selectivity. The experimental work is systematic and the results are convincing.

5) There is overlap between the paper Olfactory perception and behavioral effects of sex pheromone gland components in Helicoverpa armigera and Helicoverpa assulta Meng Xu et al. 2016 and the current paper. This overlap should be made more clear.

The content of this study is closely related with Xu et al. (2016), the previous study in our group, but there is no data overlap between them. The only data similar in the two studies is the expression level of PRs. Xu et al. (2016) used the in situ hybridization to detect the numbers of the cells expressing different PRs. In this study, we use qPCR to analyze the expression level of all the PRs. The present results from the molecular level are consistent with the previous results from the cell level.

Reviewer #2:[…] 1. I miss citation to the recent article by De Fouchier et al. in Nature Communications (DOI: 10.1038/ncomms1570910).

We accept the suggestion, and add this citation.

2. Since HarmOR14b is tuned to Z9-14:Ald, I think it's important that the authors write 1-2 sentences on how this component is a minor (but essential) sex pheromone component in H. armigera. In the current text it seems that the sex pheromone system of these two species is a two-component system with similar reverse ratios as in the two pheromone strains of Ostrinia nubilalis, while the pheromone blend of H. armigera (and also H. assulta) is a bit more complex than just a two-component blend.

We accept the suggestion, and add “(Z)-9-tetradecenal (Z9-14:Ald) acts as an antagonist in the pheromone communication of H. assulta (Boo et al., 1995; Wu et al., 2015). In that of H. armigera, Z9-14:Ald acts as an agonist in small amounts (0.3%) (Rothschild, 1978; Wu et al., 2015; Zhang et al., 2012) but an antagonist in higher amounts (1% and above) (Gothilf et al., 1978; Kehat and Dunkelblum, 1990; Wu et al., 2015).” in the revised manuscript’s Introduction.

3. The authors give dN/dS ratios (ω) for the different ORs (Results, subsection “Phylogenetic analysis of candidate PRs”), reasoning that ω < 1 indicates puryfying selection, while ω > 1 indicates positive selection. As they found a ω = 0.17 for cluster OR14b, this thus indicates puryfying selection. However, in the discussion the authors do not come back to this result and instead write "The female moth produces a pheromone blend of several components, stabilized by strong selection pressure against any change in such blends (Roelofs et al., 2002). This requires an equivalent stability from the male moths to detect the same species-specific pheromones, but at the same time should allow for a degree of plasticity to adapt to changes in pheromone structures associated with speciation." This part needs to be revised, as 'stabilized by strong selection pressure' and 'equivalent stability' are strangely used. Also, the latter part "should allow for a degree of plasticity" comes across as hand waving. Similarly, "In the course of speciation, the functional change of ORs is a gradual process with multiple amino acid mutations, a few making drastic changes and many making small modifications or even no change in function" need to be revised, as 'gradual changes' contradicts 'a few making drastic changes' and as a whole this sentence doesn't make sense.

Thanks for all the suggestions. We accept all of them and revise the manuscript as follows:

We add “The ω value for all clusters of PRs are less than 1, suggesting that all the PRs are subjected to purifying selection.” in the. In addition, we recalculate the dN/dS value of Or11 cluster, and find the correct value of Or11 cluster is less than 0.13. So the ω values of all clusters of PRs are less than 1. We have corrected the related sentences in the new version.

We revise the statement in the new version as “Under stabilizing selection, variation of the female pheromone blend is limited, and the males typically prefer the most common pheromone blends (Groot et al., 2016; Roelofs et al., 2002).” in the Discussion.

Also in the Discussion, we revise the statement in the new version as: “However, at the same time the male moths need to have a degree of plasticity to adapt to changes in signal structures associated with speciation. How mutations in PRs change their response to odorants remain unclear.”

We also revise this statement as “In the course of speciation, the functional change of ORs could be a process with multiple amino acid mutations, a few making drastic changes and many making small modifications or even no change in function.” in the Discussion.

4. Discussion, subsection “Novel identified PR and the different combinations with other PRs”, first paragraph: HassOR14b should be "HassOR16 in the same sensilla". Do the authors know where the orthologous ORs are localized in H. armigera? Is HarmOR14b also co-localized with HarmOR6 or HarmOR16? Can I deduce from the intro information that these sensilla are the type C sensilla?

To make this question clear, we add the new data about the expression pattern of all the PRs in *H. armigera* in Figure 2—figure supplement. It is clear that HassOR14b is co-localized with HassOR16 or HassOR6 in the C type sensilla in *H. assulta*. However, HarmOR14b had a very low expression level in the male antennae of *H. armigera*, even lower than HarmOr16 and HarmOr6. We speculate that the localization of PRs is not exactly same in the type C sensilla between *H. armigera* and *H. assulta*. It is still unclear about the localization of HarmOr14b, HarmOr6, and HarmOr16 in *H. armigera*. We will clarify this in future research.

5. Discussion, subsection “Novel identified PR and the different combinations with other PRs”, second paragraph:: Where are these amino acid residues located? I think it's important to specify this, especially because this is the first study (right?) where amino acid sequence changes in the intracellular domains (ICDs). I would also like to read a bit more on how the authors think these changes may alter the function. They give two very short explanations in the next paragraph, but what do they mean with 'complex deep structure'?

We used the TOPCONS (topcons.net) to predict the secondary structure of Or14b, and find that the two mutation sites were located in the intracellular domains (ICDs). This is the first study of ORs where amino acid sequence changes in the intracellular domains (ICDs) affect the tuning selectivity. We specify this in the revised Discussion. We also cite the new references for the amino acid changes may alter the function. We revised this statement as “the binding site of ligand-specific ORs, such as PRs, may have a complex structure, which involves TMDs (Leary et al., 2012), ECLs (Hughes et al., 2014) and ICDs.”

Reviewer #3:Yang et al. report that two amino acid substitutions in intracellular domains may account for the difference in ligand specificity between HarmOr14b and HassOr14b. The conclusion is based on a series of mutagenesis experiments. The results are interesting but "the story" is not ready for publication. A number of major issues are listed below. In addition, I think that the discussion of how mutations in the intracellular domains may influence ligand specificity (subsection “Two amino acids located in the intracellular domains together determine the OR selectivity”, last paragraph) is speculative and lacks both references and a possible mechanism.

Many thanks for the comments. We revised the discussion of how mutations in the intracellular domains may influence ligand specificity. We added the references in the last paragraph of the subsection “Two amino acids located in the intracellular domains together determine the OR selectivity”.

The study is mainly based on the authors´ finding that HassOR14b is responsive to Z9-16:Ald whereas HarmOR14b is responsive to Z9-14:Ald. However, according to a previous study from the same laboratory (Jiang et al., 2014) and the study by Chang et al. (2015), HassOR14b did not respond to any tested compounds including Z9-16:Ald. This needs to be clarified. The authors do not mention about this discrepancy and do not even cite the Chang et al. paper:Jiang, X.-J., Guo, H., Di, C., Yu, S., Zhu, L., Huang, L.-Q., Wang, C.-Z. (2014). Sequence similarity and functional comparisons of pheromone receptor orthologs in two closely related Helicoverpa species. Insect Biochemistry and Molecular Biology 48, 63-74.Chang, H., Guo, M., Wang, B., Liu, Y., Dong, S., and Wang, G. (2016). Sensillar expression and responses of olfactory receptors reveal different peripheral coding in two Helicoverpa species using the same pheromone components. Scientific reports, 6, 18742.

The first question is the same with the reviewer #1’s question 2. Please find our response to reviewer #1.

We discussed this discrepancy and cite the Chang et al.’s paper in the second paragraph of the subsection “Novel identified PR and the different combinations with other PRs”.

In the phylogenetic tree, the authors only included Heliothine species in OR13 cluster, but not the orthologues from other species, thus the calculation of dN/dS value is biased. Moreover, the authors say the dN/dS value of OR11 cluster is larger than 1.8, this is inconsistent with the previous findings that dN/dS value of this cluster is low (around 0.1):Zhang, D. D., and Löfstedt, C. (2013). Functional evolution of a multigene family: orthologous and paralogous pheromone receptor genes in the turnip moth, Agrotis segetum. PLoS One, 8(Fang et al., 1997), e77345.Zhang, Y. N., Zhang, J., Yan, S. W., Chang, H. T., Liu, Y., Wang, G. R., and Dong, S. L. (2014). Functional characterization of sex pheromone receptors in the purple stem borer, Sesamia inferens (Walker). Insect molecular biology, 23(Carraher et al., 2015), 611-620.

We accepted the suggestions and have recalculated the dN/dS value in Or13 cluster including the orthologues from other species, and the ω value now is less than 0.17. We also rebuilt the Figure 1 in the revised version.

Thank you for this important comment. We recalculated the dN/dS value of Or11 cluster, and found the value we previously calculated about Or11 is not correct. In the sequence alignment analysis, we wrongly used “Align by ClustalW” (It should be “Align by ClustalW (Codons)”). We are very sorry for this mistake. Now, the value of Or11 cluster corrected is less than 0.13, which is consistent with Zhang and Löfstedt (2013) and Zhang et al. (2014). We have corrected this mistake and double checked all the calculated ω values in the revised manuscript in Figure 1 and subsection “Phylogenetic analysis of candidate PRs”.

The in situ hybridization images are not convincing and this is problematic as the results do not correspond to what was reported in Chang et al. (2015). These authors claimed that HassOR6 and HassOR16 are localised in the same sensillum (Chang et al., 2015).

We provide more in situ hybridization images of our experiments in Author response Figure 1, Figure 2 and Figure 3). For each treatment, we present three more sets of images besides the images we present in the manuscript (Figure 5). The results are clear and repeatable.

Author response image 1 – Part 1, HassOr14b and HassOr6 are co-localized in some sensilla (arrows); Part 2: Only HassOr14b is detected in other sensilla (arrows).

Author response image 2 – Part 1, HassOr14b and HassOr16 are co-localized in some sensilla (arrows); Part 2, only HassOr14b is detected in other sensilla (arrows).

Author response image 3 – HassOr6 and HassOr16 are always expressed in different sensilla. Part 1, HassOr16 is only expressed in some sensilla (arrows); Part 2, HassOr6 is only expressed in some sensilla (arrows).

All these indicate that HassOr14b was co-localized with HassOr6 or HassOr16 in the sensilla, and the results match with the functions of the PRs, the expression patterns of PRs, and the electrophysiology of related OSNs in *H. assulta* (Chang et al., 2016; Xu et al., 2016).

**Author response image 1. respfig1:** Two-colour *in situ* hybridization visualizing the combinations of HassOr14b and HassOr6 in male antennae of *H. assulta*. Arrows indicate the cell location. Scale bars: 20 μm.

**Author response image 2. respfig2:** Two-colour in situ hybridization visualizing the combinations of HassOr14b and HassOr16 in male antennae of *H. assulta*. Arrows indicate the cell location. Scale bars: 20 μm.

**Author response image 3. respfig3:** Two-colour in situ hybridization visualizing the combinations of HassOr16 and HassOr6 in male antennae of *H. assulta*. Arrows indicate the cell location. Scale bars: 20 μm.

In the paper of Chang et al.(2016), the only set of images about the localization of HassOr6 and HassOr16 in *H. assulta* in Figure 8 of Chang et al.is as the following (Author response image 4). From the images, it is very difficult to judge OR6 and Or16 are co-localized.

**Author response image 4. respfig4:** Two-colour in situ hybridization visualizing the combinations of HassOr6 and HassOr16 in male antennae of *H*. *assulta*. Arrows indicate the cell location. Scale bar: 20 μm; the boxed area: 10 μm (adapted from Figure 8 Chang et al., 2016).

Chang et al.(2016) suggested that HassOr6 is the receptor of Z9-16:Ald although they confirmed that the most effective ligand of HassOr6 is Z9-16:OH in *Xenopus* system (Supplementary Figure 1). From the *Xenopus* result (Z9-16:Ald is not the most effective ligand of HassOr6.) to the expression pattern of PRs (HassOr6 is not the highest expressed PR in male antennae of *H. assulta*.) and the electrophysiological data (the OSNs responding to Z9-16:Ald is the most abundant in male antennae.), all the evidences do not support that HassOr6 is the receptor of Z9-16:Ald.Discussion subsection “Novel identified PR and the different combinations with other PRs”, first paragraph: 'This suggests that the ancestor of the two sister species not only changed OR14b's expressing level, but also altered its tuning selectivity to meet the species specific demands': The authors did not compare the expression levels of HassOR14b and HarmOR14b so the statement is not supported. In addition, it is not clear to me how the authors can conclude anything about the expression levels in the ancestor. Based on analysis of the contemporary species (which at some point had a common ancestor) we can at the best conclude that expression levels are different in these two species. The reasoning further involves a teleological argument, i.e. that the species that evolved from the common ancestor had some "specific demands". This is not how evolution works.

Thank you for these critical comments and sorry for our poor expression in English. We add the data about the expression pattern of all the PRs in *H. armigera* in Figure 2—figure supplement 2. It is clear that HarmOR14b had a very low expression level in the male antennae, even lower than HarmOr16 and HarmOr6. As HassOr14b is most highly expressed in *H. assulta*, and the functional study confirms that HassOR14b is specifically tuned to Z9-16:Ald, while its ortholog HarmOr14b is specifically tuned to Z9-14:Ald, we suggest that the Or14b orthologs of the two sister species not only changed Or14b’s expressing level, but also altered its tuning selectivity in evolution.

We accept the suggestion and revised the sentence as “This suggests that the two sister species not only changed Or14b’s expressing level, but also altered its tuning selectivity in speciation.”.

In the subsection “Quantitative real-time PCR”: the authors write that the reference gene is 18s rRNA, but the GenBank number provided is actually the actin gene. This is confusing.

We are sorry for this mistake and thank you for your carefulness. We used the 18s rRNA as the reference gene, and its GenBank number is changed into EU057177.1 accordingly. We have corrected it in the subsection “Quantitative real-time PCR”.

In the subsection “Construction of the mutation sequences”: The description of the construction strategy would benefit from a diagram visualizing the different steps.

It is a very nice suggestion, we have added the diagram visualizing the construction strategy in the revised manuscript (Figure 7—figure supplement 1).

[Editors' note: further revisions were requested prior to acceptance, as described below.]

The reviewers have appreciated your effective revision of the manuscript and the extensive explanation in the rebuttal. They conclude that your study provides very interesting results on how small differences in pheromone receptors may influence ligand selectivity. Still some important issues remain and I invite you to address these comments by the reviewers and prepare a second revision of the manuscript. Specific attention should be paid to sentences where you refer to the literature on odour reception and genetic mechanisms and evolutionary consequences. Some of these sentences are highlighted in the reviews, while others are not. For example, what is meant with 'a few steps' in the last sentence of the Abstract.

Thank you so much for your comments to the last revision and giving us a chance to revise the manuscript again. Although we cannot identify the highlights in the reviews, we have paid attention to the related sentences and rephrased them. The last sentence of the Abstract is revised as follows: “We conclude that species-specific changes in the tuning specificity of the PRs in the two *Helicoverpa* moth species could be achieved with just a few amino acid substitutions, which provides new insights into the evolution of closely related moth species.”.

Reviewer #1:[…] The authors have made useful changes but I think a little more discussion of some of the issues with the Xenopus system in this specific case is warranted because the same issue is likely to arise in future studies.

Thank you so much for giving your recognition to our revision. We added some sentences in Discussion to discuss the issues with the *Xenopus* system: “It is worth noting that inward currents of the oocytes expressing HassOr14b induced by Z9-16:Ald were distinct but relatively low. […] A clear dose response curve to the most effective ligand is always helpful to confirm the receptor’s function.”.

I realize that in the current Abstract and in most of the manuscript the authors avoid discussing the relevance of their findings to the evolution of the two species of Helicoverpa. For example, in the Abstract they state that "We conclude that species-specific changes in the tuning specificity of the PRs of male moths could be achieved with just a few steps". As long as the authors don't indicate that their findings help us to specifically understand the evolution of the two Helicoverpa moths, I think they are on solid ground. The two mutations they study certainly change the tuning specificity.

Thank you so much for your suggestions. We added the statement in the revised Abstract as “We conclude that species-specific changes in the tuning specificity of the PRs in the two *Helicoverpa* moth species could be achieved with just a few amino acid substitutions, which provides new insights into the evolution of closely related moth species.”. We added the statement in the revised Discussion as “The peripheral modifications of the two closely related species took place in both PR expression level and PR tuning selectivity. These findings not only help us specifically understand the evolution of the two *Helicoverpa* species, but also provide new insights into the structure and function of cell membrane receptors.”.

It still would be good if the authors could elaborate on the fact their findings are of most interest in terms of neurophysiology and are not a direct commentary on the evolution of these specific moths.

Thank you for your suggestions. This is a good point. We added one paragraph in Discussion as follows: “Animal nervous systems are shaped by shifting environmental selection pressures to perceive and respond to new sensory cues (Prieto-Godino et al., 2017). […] How mutations in olfactory receptors change the olfactory responses of animals and eventually impact on the evolution of animal behavior is crucial but remains unclear.”

Reviewer #2:.[…] 1) In Figure 3 the authors show that HasOR14b responds to Z9-16:Ald, but when comparing the different graphs, the y-axis in Figure 3 is in a different (smaller) scale than the y-axis in Figure 3—figure supplement 1, which shows that HassOr6 actually responds more to Z9-16:Ald (300 nA) than HassOr14b (70 nA). Such a difference in scaling is also present in Figure 8; HassOr14b shows a current response to Z9-16:Ald of 70 nA again, and/but HarmOr14b shows a current response of 30-40 (?) nA. I do understand that HarmOr14b mostly responds to Z9-14:Ald, but the response to Z9-16:Ald may also be important, no? (In this respect it is indeed very impressive that the authors found an almost 10x stronger response when mutating T335I (see Figure 7—figure supplement 2). Which relates to my next question:

The *Xenopus* system is a heterologous expression system. Its main disadvantage is that the oocytes are not the native cells in which the PRs are normally expressed. Although the functional properties that are observed may not be identical to those characterized in native olfactory sensory neurons, the most effective ligand obtained by using the *Xenopus* system is convincing. However, the response range is often wider than the natural situation (Wang et al., 2016). It is hard to judge whether the compounds inducing lower currents could actually activate the related neurons in the sensilla. For example, the neurons expressing HarmOr13 and HassOr13 in Type A sensilla are only tuning to Z11-16:Ald specifically (Wu et al., 2013; Chang et al., 2016; Xu et al., 2016). When HarmOr13 and HassOr13 are expressed in *Xenopus oocytes*, the most effective ligand is still Z11-16:Ald, but they also show a relatively lower response to Z9-14:Ald (Jiang et al., 2014; Liu et al., 2013). So we would better pay more attention to the response selectivity rather than the absolute current value among different ORs. In the Results part, we revise some statements in the paragraphs of “Regional mutations of HassOr14b and functional analysis” and “Site-specific mutations of HassOr14b and functional analysis” (first paragraph).

Thank you for your comment on the results. We think “Figure 7—figure supplement 2” you mentioned is “Figure 8” because the result of T355I is in Figure 8.

2) As Z9-16:Ald is also an important (albeit minor) sex pheromone component of H. armigera, with what receptor(s) do the authors think H. armigera is perceiving this component?

This is a very good question. The receptor(s) tuning to Z9-16:Ald of *H. armigera* is still a mystery.

In the PR clade in phylogenetic analysis, 7 PRs have been identified, they are HarmOr6, HarmOr11, HarmOr13, HarmOr14, HarmOr14b, HarmOr15, and HarmOr16. Up to now, 4 of them are functional characterized: HarmOr6, HarmOr13, HarmOr14b, HarmOr16. The most effective ligands are Z9-16:OH for HarmOr6 (Jiang et al., 2014), Z11-16:Ald for HarmOr13 (Liu et al., 2013), Z9-14:Ald for HarmOr14b (Jiang et al., 2014, and this study), Z11-16:OH (Chang et al., 2017; Liu et al., 2013) and Z9-14:Ald (Liu et al., 2013) for HarmOr16. The function of other 3 PRs (HarmOr11, HarmOr14, HarmOr15) are not yet identified. HarmOr11 is co-localized with HarmOr13 in Type A sensilla (Chang et al., 2016), it has no possibility tuning to Z9-16:Ald; HarmOr15 has a much lower expression in male *H. armigera* (this study, Figure 2—figure supplement 2), so could not be the candidate receptor. Therefore, it could be HarmOr14 as the potential receptor tuning to Z9-16:Ald, although all the studies up to now failed to identify its function (Jiang et al., 2014; Liu et al., 2013). In the recent ESITO meeting, a new PR clade was reported in *Spodoptera*, it would be also interesting to explore the new PRs in *Helicoverpa armigera* in the further study.

3) In Figure 2 the authors show that HassOR6 is highly expressed in males, and a little in females. Which makes sense to me, as females probably smell their own pheromone as well. I don't think that sex-specific expression is a necessary criterium for an OR to be a sex pheromone OR.

We totally agree with this. A recent study on *Heliothis virescens* (Zielonka et al., 2016) find that HR6 is expressed in both female and male antennae, which indicates that the females can detect pheromone components released by themselves or by conspecifics. We also agree that the sex-specific expression is not a necessary criterium for an OR to be a PR, functional analysis is the only criterium to judge whether it is a PR. However, the male biased expression pattern could narrow the scope of the candidate PRs in moths, which is the efficient way to identify most of the PRs.

4) The authors have responded to previous comments on why previous functional assays with HassOr14b didn't work by writing that the sequence had not been correct in previous work and that this is now corrected. In Figure 3—figure supplement 3 they specify which 3 amino acids have been corrected (Yang et al. is the current sequence used). In checking these 3 amino acids in Figure 6 found them in RIII, in RVII and in RVIII. I find it hard to determine whether one of these 3 SNPs were among the ones tested in Figure 7—figure supplement 2. Still, I think the authors make a convincing case that the right sequence together with the selection of the appropriate expression vector is crucial to identify the function of receptors.

In the site mutation analysis, we just focused on the different amino acids between orthologs of *H. assulta* to *H. armigera.* Among the three revised amino acids of HassOr14b, two of them located in RIII and RVIII respectively are the same with those of HarmOr14b. The third amino acid located in RVII of HassOr14b is different from that of HarmOr14b, when we replaced the whole region VII of HassOr14b with that of HarmOr14b, we did not find the selectivity change. Therefore, we did not test these three amino acids specifically. We think “Figure 7—figure supplement 2” you mentioned should be “Figure 8”. Thank you for your comments to our results.

5) Hopefully Figure 7—figure supplement 2 is not a supplementary figure, as I think this figure is a great way of showing the results of the single mutations.

We think that “Figure 7—figure supplement 2” you mentioned is “Figure 8”, which is the result of the two-electrode voltage-clamp recordings of site mutations with pheromone components and analogs. This result is not a supplementary figure. Thank you for your comments to our results.

6) As for the in situ hybridization pictures, these are clear to me, but I leave this to the more experienced reviewer(s) to decide whether this is sufficiently clear.

Thank you for your comments to our results.

Reviewer #3:[…] In the rebuttal letter the authors write: 'Basically there are two major concerns: (Baker et al., 2004) the suitability of the Xenopus system.… (Boo et al., 1995) Contradiction of our in situ hybridization results'In my opinion this is not correct. At least not when it comes to my concerns. My major concern was that the authors avoided mentioning the inconsistency in the response of wild type HassOR14b compared to previous publications. In the current version, the authors explain that the gene sequences were not correct in previous publications. However, the explanation is not straight forward:

The full-length sequences of HassOr14b and HarmOr14b were first cloned by Jiang et al. in 2014, and Jiang et al. identified HarmOr14b was tuned to Z9-14:Ald, but failed to identify the function of HassOr14b (Jiang et al., 2014). In 2016, Chang et al. proved the function of HarmOr14b, and tried to identify the function of HassOr14b again but also failed. Because Jiang et al. (2014) is the first study of HassOr14b and there is no new progress of this gene in functional analysis in the following study, we just cited Jiang et al.’s work in the very first manuscript version. From the last revision, we accepted the reviewer’s suggestion and cited both of them.

- The sequences in Chang et al., 2016 were based on the transcriptome data from the previous paper (Zhang et al., 2015) from the same group. There's a single amino acid difference of HassOR14b sequence between the current manuscript and the sequence in Chang's paper. Could it simply be explained by the use of different polymerases? How could the authors exclude the possibility of polymorphism? Although possible, it's not very likely that this one amino acid difference would totally abolish the response to Z9-16:Ald (No response to Z9-16:Ald or any other ligand in Chang et al. 2016) but significant response to Z9-16:Ald in the current study (70 nA – Figure 8). The authors of the present manuscript can of course not take the responsibility of any possible flaws in a study published by another group but the discrepancy requires an explicit comment in the Discussion.Zhang, J., Wang, B., Dong, S., Cao, D., Dong, J., Walker, W. B.,.… & Wang, G. (2015). Antennal transcriptome analysis and comparison of chemosensory gene families in two closely related noctuidae moths, Helicoverpa armigera and H. assulta. PloS one, 10(Boo et al., 1995), e0117054.

Many factors are possible to result in the sequence cloning error. We think DNA polymerase is the key factor in PCR systems. Among the DNA polymerases supplied by many companies, the fidelity of them is quite different. By our experiences, in the purpose of cloning the sequences for functional analysis, the high fidelity enzyme should be the first choice. The DNA polymerase used in Chang et al.’s study is LA-*Taq* (Chang et al., 2016), which is not a high fidelity enzyme.

Concerning the possibility of polymorphism, we excluded it using the following strategy: The transcriptome data represents many separate individuals, and the results of sequencing include abundant separate “reads”. For each gene, there are many “reads” mapping to it, but each “reads” is not exactly the same in all the positions, which may come from the polymorphism in such position. If all of the “reads” in one position is the same, we can exclude the possibility of polymorphism in that position. We analyzed all the related transcriptome data of *H. assulta* carefully, and if we found that all of the “reads” in one position is the same, we exclude the possibility of polymorphism in that position.

Concerning the amino acid difference from that in Chang et al. (2016), this amino acid is located in RI of HassOr14b, which is the same with that of HarmOr14b, so we do not test this amino acid specifically because in the site mutation experiments we just focused on the different amino acids between orthologs of *H. assulta* to *H. armigera.* Although we do not test whether the change of this amino acid would totally abolish the response of HassOr14b, we have experience that one amino acid change in HarmOr13 could totally abolish its function (our unpublished data). Therefore, we suggest that the accuracy and integrity of the sequence is crucial to identify the function of ORs. We have explained this in the second paragraph of the Discussion subsection “Novel identified PR and the different combinations with other PRs”.

I do not understand how different expression vectors could cause the difference in sequence. The authors have to explain how. Otherwise it is just an ad hoc explanation that needs to be tested. It's the cRNAs that are injected and expressed in the oocytes. The resulting receptor proteins should have the same sequences despite of what the original expression vectors were. In addition, the pT7TS vectors have been widely and successfully used in oocyte recordings and OR studies. Why should this vector not be suitable for HassOR14b expression and oocyte recordings?

It is true that the vector would not cause the difference in sequence, but the vector itself used in the *Xenopus* system could affect the identification of the ORs. pT7TS vectors have been successfully used in oocyte recordings in many OR studies, but Chang et al. used it and failed to identify HassOr14b’s function. We use the pCS2+ vector and successfully identify HassOr14b. Comparing with the Chang et al.’s study, we notice that there are two major differences: one is on the sequence of HassOr14b, another is on the expression vector, so we speculate that these two differences could be the possible reasons.

The authors write in the rebuttal and mention in the Discussion that they re-sequenced HassOr14b and came up with the "right sequence". This is a new (and important) result and as such it should be mentioned in the Results section and the relevant methods and materials should be described in Materials and methods before the discrepancy with previous studies is discussed in the Discussion section.

Thank you and we accept the suggestion, and added this information in the Results subsection “PR specifically tuned to Z9-16:Ald in *H. assulta*”, and the Materials and methods subsection “Cloning of the candidate pheromone receptor of *H. assulta* and *H. armigera*”.

As far as I understand reviewer #1 was not really questioning oocytes as a suitable system for OR de-orphanization in general, but said that the system may not be efficient for testing HassOR14b. In my opinion, however, a current of 60 nA is not too small to be noticed. Hence this is not the reason why Chang et al. did not find a response, but there seems to be a real difference in activity between the two studies.It remains mysterious to me why the wild type HassOR14b responds so differently in this and previous studies and it's problematic that the authors did not mention this inconsistency in the first version. In addition, a number of other mistakes/errors were pointed out by me and the other reviewers. The authors have now corrected these errors but can we be sure that additional errors did not go unnoticed by the reviewers? Careful doublechecking appears necessary.

Thank you for explaining the results in the *Xenopus* oocyte system. Concerning the difference in activity, the fact is not that the previous studies found one function and we find another function of HassOr14b, but is that the previous studies all failed to identify its function and we find it in this study. We suggest that it would be the use of the right sequence and the selection of the expression vector that result in our success in this study. As both Jiang et al. and Chang et al. failed to identify the function of HassOr14b, and Jiang et al. is the first study identifying HassOr14b (Jiang et al., 2014, Chang et al., 2016), we just cited Jiang et al.’s work in the very first manuscript version. In the revised manuscript, we accept your suggestion and cite both of them. Thanks for your suggestion and we had double-checked the manuscript carefully.

Some other points for your consideration.1) 'Unlike general odorant receptors (ORs) that typically bind more than one ligand (de Fouchier et al., 2017), PRs are narrowly tuned to specific pheromone components (Grosse-Wilde et al., 2007).' This statement is not correct. There are both narrowly tuned PRs and broadly tuned PRs reported in the literature (Miura et al., 2010; Wanner et al. 2010; Zhang and Löfstedt 2015). Even in Grosse-Wilde et al., 2007 cited by the authors, it was proposed that 'there are two different designs of pheromone receptors'. The authors have to be more precise in their statements and more rigorous when citing the literature:Miura, N., Nakagawa, T., Touhara, K., and Ishikawa, Y. (2010). Broadly and narrowly tuned odorant receptors are involved in female sex pheromone reception in Ostrinia moths. Insect biochemistry and molecular biology, 40(Baker et al., 2004), 64-73.Wanner, K. W., Nichols, A. S., Allen, J. E., Bunger, P. L., Garczynski, S. F., Linn Jr, C. E., […] and Luetje, C. W. (2010). Sex pheromone receptor specificity in the European corn borer moth, Ostrinia nubilalis. PLoS One, 5(Baker et al., 2004), e8685.Zhang, D. D., and Löfstedt, C. (2015). Moth pheromone receptors: gene sequences, function, and evolution. Frontiers in Ecology and Evolution, 3, 105.

We accept the suggestion, and revised the statement as “PRs are in general narrowly tuned to specific pheromone components (Grosse-Wilde et al., 2007; Miura et al., 2010; Zhang and Löfstedt, 2015).”

2) 'For example, BmOr1 of.[…] to 1,3Z,6Z,9Z-19:H (Zhang et al., 2016).' As we suggested before, there's no point of listing these examples so this sentence can be removed.

We accept the suggestion and removed this sentence in the revised manuscript.

3) Results and Discussion sections: 'all the PRs and Orcos are subjected to purifying selection which is consistent with the previous studies (Zhang and Löfstedt, 2013; Zhang et al., 2014b)'. This is not a correct reference. Zhang and Löfstedt (2013) as well as other researchers (e.g. Leary et al., 2012) found that some PR clusters are under purifying selection while some are under relatively relaxed selective pressure. Since the authors used a limited number of moth species in the phylogenetic tree in the current study, it's too daring to draw the conclusion that 'all the PRs and Orcos are subjected to purifying selection'. Leary, G. P., Allen, J. E., Bunger, P. L., Luginbill, J. B., Linn, C. E., Macallister, I. E.,.[…] and Wanner, K. W. (2012). Single mutation to a sex pheromone receptor provides adaptive specificity between closely related moth species. Proceedings of the National Academy of Sciences, 109(Liu et al., 2014), 14081-14086.

We accept the suggestion and revised them as “this indicates that all the PRs and Orcos analyzed in this study are subjected to purifying selection, which is consistent with the previous studies (Zhang and Löfstedt, 2013; Zhang et al., 2014b).”; and “The ω value for all clusters of PRs analyzed in this study are less than 1, suggesting that PRs would be subjected to purifying selection.”

4) In situ results: 'in some sensilla only HassOr14b was detected'. For the Discussion: Is this consistent with previous SSR data? Which gene is supposed to be expressed in the neighboring neuron?

We found that HassOr14b and HassOr6 were co-localized in some sensilla, while in other sensilla only HassOr14b was detected. A similar situation was observed for HassOr14b and HassOr16. They were co-localized in some sensilla, but only HassOr14b was detected in other sensilla. HassOr6 and HassOr16 were always expressed in different sensilla. These results indicate that HassOr6 or HassOr16 were co-localized with HassOr14b in the neighboring neurons in the Type C sensilla, which is consistent with the previous SSR data that there are subtypes in the type C sensilla. We revised the statement in the Discussion as “HassOr14b is co-localized with HassOR6 or HassOR16 in the neighboring neurons in the same sensilla,”, and added the statement as “Our results indicate that there are different combinations of the PRs in the C type sensilla, which is consistent with the previous single sensillum recording results that there are subtypes in the type C sensilla (Xu et al., 2016).”

5) The authors refer to the transcriptome a couple of times in the manuscript. Have the authors deposited the transcriptome data in a public database?

The transcriptome data of *H. assulta* and *H. armigera* submitted by MPICE is deposited in the NCBI-Sequence Read Archive (SRA) (https://www.ncbi.nlm.nih.gov/sra). The transcriptome data of *H. assulta* and *H. armigera* acquired by our group is still under analysis (our unpublished work), and we will deposit it in a public database when we finish the analysis.

6) Discussion subsection “Implications for the modulation and evolution of OR selectivity”: 'The olfactory systems found in all animals have nearly the same design feathers, which give olfaction a considerably flexibility for signaling to evolve.' I don't understand this sentence. What does 'design feathers' mean? Should it be "features"?

We are sorry for this mistake and thank you for your carefulness. It should be “features” and we have corrected it in the first paragraph of the subsection “Implications for the modulation and evolution of OR selectivity”.

7) The English still needs some attention throughout the manuscript before its scientific merits can be definitely evaluated.

We accept the suggestion, and have revised the language throughout the manuscript.